# Two-Player Zero-Sum Differential Games with One-Sided Information and Continuous Actions

## Abstract

Unlike Poker where the action space $\mathcal{A}$ is discrete, differential games in the physical world often have continuous action spaces not amenable to discrete abstraction, rendering no-regret algorithms with $\mathcal{O}(|\mathcal{A}|)$ complexity not scalable. To address this challenge within the scope of two-player zero-sum (2p0s) games with one-sided information, we show that (1) a computational complexity independent of $|\mathcal{A}|$ can be achieved by exploiting the "Cav u" property of behavioral strategies in incomplete-information games and the Isaacs' condition that commonly holds for control systems, and that (2) the computation of the two equilibrium strategies can be decoupled under the Isaacs' condition. We provide computational complexity of the resultant algorithm for approximating continuous-action mixed strategies (CAMS). Empirically, we demonstrate correctness of CAMS using a homing game where the Nash equilibrium exists analytically, and scalability through the same game with higher-dimensional actions. Codes available in anonymous repo.

## 1 Introduction

The strength of game solvers has grown rapidly in the last decade, beating elite-level human players in Chess (Silver et al., 2017a), Go (Silver et al., 2017b), Poker (Brown & Sandholm, 2019; Brown et al., 2020b), Diplomacy (, FAIR), Stratego (Perolat et al., 2022), among others with increasing complexity. These successes motivated recent interests in solving differential games in continuous time and space, e.g., competitive sports (Wang et al., 2024; Ghimire et al., 2024), where critical strategic plays should be executed precisely within the continuous action space and at specific moments in time (e.g., consider set piece scenarios in soccer). However, existing no-regret solvers, e.g., CFR+ (Tammelin, 2014) and its variants (Burch et al., 2014; Moravčík et al., 2017; Brown et al., 2020b; Lanctot et al., 2009), and last-iterate online learning algorithms, e.g., variants of follow the regularized leader (FTRL) (McMahan, 2011; Perolat et al., 2021) and of mirror descent (Sokota et al., 2022; Cen et al., 2021; Vieillard et al., 2020), are designed for discrete actions and have computational complexities increasing with respect to the size of the action space $\mathcal{A}$. Thus applying these algorithms to differential games would require either insightful action and time abstraction or enormous compute, neither of which are readily available.

As a one step towards addressing this challenge, our study focuses on games with one-sided information, which represent a variety of attack-defence scenarios: Both players have common knowledge about the finite set of $I$ possible payoff types and nature's distribution over these types $p_0$. At the beginning of the game, nature draws a type and informs Player 1 (P1) about the type but not P2. As the game progresses, the public belief about the chosen type is updated from $p_0$ based on the action sequence taken by P1 via the Bayes rule. P1's goal is to maximize the expected payoff over $p_0$. This game is proved to have a value under Isaacs' condition (Cardaliaguet, 2009). Due to the zero-sum nature, P1 may need to delay information release or manipulate P2's belief to take full advantage of information asymmetry; and P2's strategy is to optimize the worst-case payoff. Real-world examples of the game include man-on-man matchup in sports where the attacker has private information on which play is to be executed, and missile defense where multiple potential targets are concerned.

The two differences between our game and commonly studied imperfect-information extensive-form games (IIEFGs) (Sandholm, 2010; Perolat et al., 2022; , FAIR) are that: (1) IIEFGs often have belief

spaces (e.g., belief about opponent's hands in Poker) larger than their abstracted action spaces (e.g., betting categories in Poker), and (2) information asymmetry in our games is only one-sided. This paper investigates the potential computational advantages from exploiting these differences via the following insights: (1) At any infostate, P1's (resp. P2's) behavioral strategy is $I$ (resp. $I + 1$)-atomic and convexifies the primal (resp. dual) value with respect to the public belief. With this, we can reformulate the convex-concave minimax problem of size $\mathcal{O}(|\mathcal{A}|)$ at each infostate into a nonconvex-nonconcave problem of size $\mathcal{O}(I^2)$. When $I^2 \ll |\mathcal{A}|$, and in particular when $|\mathcal{A}| = \infty$, the latter of a much smaller scale becomes more efficient to solve in practice. And (2) due to the one-sidedness of information, the equilibrium behavioral strategies of P1 and P2 can be solved separately through primal and dual formulations of the game. In each formulation, the opponent plays pure best responses under Isaacs' condition. This formulation avoids recurrent learning dynamics between the pair of strategies without regularization (Perolat et al., 2021).

Our contributions are summarized as follows:

- We show that when the imperfect information of a 2p0s differential game is only about the payoff type and is one-sided, and when Isaacs' condition (see equation 2) holds, the game can be solved with a computational complexity independent of the size of the action space. To our best knowledge, this is the first time the utility of these properties in jointly addressing the scalability challenge is explained in the context of imperfect-information games.
- We propose a continuous-action mixed strategy (CAMS) algorithm that demonstrates superior strategy approximation accuracy and computational cost against SOTA game solvers including CFR+ (Tammelin, 2014), MMD (Sokota et al., 2022), and Deep-CFR (Brown et al., 2019). CAMS also approximates reasonable strategies for games that are intractable for IIEFG solvers.

## 2 RELATED WORK

**2p0s games with incomplete information.**   Harsanyi (1967) introduced a Bayesian game framework to solve incomplete-information normal-form games by transforming the game into an imperfect-information one involving a chance mechanism. The seminal work of Aumann et al. (1995) extended this idea to repeated games and established the connection between value convexification and belief manipulation. Within the same framework, Blackwell's approachability theorem (Blackwell, 1956) naturally becomes the theoretical support for the optimal strategy of the uninformed player (P2). Building on top of Aumann et al. (1995), De Meyer (1996) introduced the concept of a dual game in which the behavioral strategy of the uninformed player becomes Markov. This concept later helped Cardaliaguet (2007); Ghimire et al. (2024) to establish the value existence proof for 2p0s differential games with incomplete information. Unlike repeated games in which belief manipulation occurs only in the first round of the game, differential games may have multiple critical collocation points in the joint space of time, state, and public belief where belief manipulations are necessary to achieve Nash, depending on the specifications of system dynamics, payoffs, and state constraints (Ghimire et al., 2024). For this reason, scalable value and strategy approximation for 2p0s differential games with incomplete information has not yet been achieved.

**Imperfect information extensive-form games.**   IIEFGs represent the more general set of simultaneous or sequential multi-agent decision-making problems with finite horizons. Since any 2p0s IIEFG with finite action sets has a normal-form formulation, a unique Nash equilibrium always exists in the space of mixed strategies. Significant efforts have been taken to find equilibrium of large IIEFGs such as poker (Koller & Megiddo, 1992; Billings et al., 2003; Gilpin & Sandholm, 2006; Gilpin et al., 2007; Sandholm, 2010; Brown & Sandholm, 2019), with a converging set of algorithms that are no-regret, average- or last-iterate converging, and with sublinear or linear convergence rates (Zinkevich et al., 2007; Abernethy et al., 2011; McMahan, 2011; Tammelin, 2014; Johanson et al., 2012; Lanctot et al., 2009; Brown et al., 2019; 2020a; Perolat et al., 2021; Sokota et al., 2022; Perolat et al., 2022; Schmid et al., 2023) (see summary in Tab. 1). These algorithms all have computational complexities increasing with $|\mathcal{A}|$, provided that the equilibrium behavioral strategy lies in the interior of the simplex $\Delta(|\mathcal{A}|)$ (see discussion in Appendix C). Critically, this assumption does not hold for differential games equipped with the Isaacs' condition, in which case the equilibrium strategy is mostly pure along the game tree, and is atomic on $\mathcal{A}$ when mixed, as we will elaborate in Sec. 3.

Table 1: Solver computational complexity with respect to action space $\mathcal{A}$ and equilibrium error $\varepsilon$

| Algorithm | Complexity |
|---|---|
| CFR variants (Zinkevich et al., 2007; Lanctot et al., 2009; Brown et al., 2019; Tammelin, 2014; Johanson et al., 2012) | $\mathcal{O}(|\mathcal{A}|\varepsilon^{-2})$ to $\varepsilon$-Nash |
| FTRL variants & MMD  (McMahan, 2011; Perolat et al., 2021) | $\mathcal{O}(\ln(|\mathcal{A}|\varepsilon^{-1}))$ to $\varepsilon$-QRE |

**Descent-ascent algorithms for nonconvex-nonconcave minimax problems.** Existing developments in IIEFGs focused on convex-concave minimax problems due to the bilinear form of the expected payoff through the conversion of games to their normal forms. This paper, on the other hand, investigates the nonconvex-nonconcave minimax problems to be solved at every infostate when actions are considered continuous. To this end, we use the doubly smoothed gradient descent ascent method (DS-GDA) which has a worst-case complexity of $\mathcal{O}(\varepsilon^{-4})$ (Zheng et al., 2023).

## 3 ZERO-SUM DIFFERENTIAL GAMES WITH ONE-SIDED INFORMATION

**Notations and preliminaries.** We use $\Delta(I)$ as the simplex in $\mathbb{R}^I$, $[T] := [1, ..., T]$, $a[i]$ as the $i$th element of vector $a$, $\partial V$ as the subgradient of function $V$, and $< \cdot, \cdot >$ for vector product. Consider a time-invariant dynamical system that defines the evolution of the joint state $x \in \mathcal{X} \subseteq \mathbb{R}^{d_x}$ of P1 and P2 with control inputs $u \in \mathcal{U}$ and $v \in \mathcal{V}$, respectively:

$$\dot{x}(t) = f(x(t), u, v). \tag{1}$$

The game starts at $t_0 \in [0, T]$ from some initial state $x(t_0) = x_0$. The initial belief $p_0 \in \Delta(I)$ is set to nature's distribution. P1 of type $i$ accumulates a running cost $l_i(u, v)$ during the game and receives a terminal cost $g_i(x(T))$, where $i \sim p_0$. We introduce the following assumptions under which the game has a value (Cardaliaguet, 2009):

1. $\mathcal{U} \subseteq \mathbb{R}^{d_u}$ and $\mathcal{V} \subseteq \mathbb{R}^{d_v}$ are compact and finite-dimensional sets.

2. $f : \mathcal{X} \times \mathcal{U} \times \mathcal{V} \to \mathcal{X}$ is bounded, continuous, and uniformly Lipschitz continuous with respect to $x$.

3. $g_i : \mathcal{X} \to \mathbb{R}$ and $l_i : \mathcal{U} \times \mathcal{V} \to \mathbb{R}$ are Lipschitz continuous and bounded.

4. Isaacs' condition holds for the Hamiltonian $H : \mathcal{X} \times \mathbb{R}^{d_x} \to \mathbb{R}$:

$$H(x, \xi) := \min_{u \in \mathcal{U}} \max_{v \in \mathcal{V}} f(x, u, v)^\top \xi - l_i(u, v) = \max_{v \in \mathcal{V}} \min_{u \in \mathcal{U}} f(x, u, v)^\top \xi - l_i(u, v). \tag{2}$$

5. Both players have full knowledge about $f$, $\{g_i\}_{i=1}^I$, $\{l_i\}_{i=1}^I$, $p_0$, and the Nash equilibrium of the game. Control inputs and states are fully observable and we assume perfect recall.

Critically, the Isaacs' condition allows any complete-information 2p0s games between P1 and P2 to have pure Nash.

Denote by $\{\mathcal{H}_r^i(t)\}_{i=1}^I$ the joint set of behavioral strategies of P1, and $\mathcal{Z}_r(t)$ the set of behavioral strategies of P2. The subscript "r" emphasizes the probabilistic nature of behavioral strategies. $X_T^{t_0, x_0, \eta_i, \zeta}$ is the random final state arrived from $x_0$ following the strategy pair $(\eta_i, \zeta)$ and by solving equation 1. Note that P1 chooses his strategy $\eta_i \in \mathcal{H}_r^i(t)$ according to his type $i$, yet P2's strategy $\zeta$ is independent of $i$. We consider strategies that are non-anticipative with delay (see (Cardaliaguet, 2007)). Practically, P1's strategy at any state $(t, x, p) \in [0, T] \times \mathcal{X} \times \Delta(I)$ contains $I$ probability measures over $\mathcal{U}$, and P2's is a probability measure over $\mathcal{V}$. Let $(\Omega_\eta, \mathcal{F}_\eta, \mathbf{P}_\eta)$ be the probability space associated with $\mathcal{H}_r^i(t)$ for $i \in [I]$ and $t \in [0, T]$, and similarly define $(\Omega_\zeta, \mathcal{F}_\zeta, \mathbf{P}_\zeta)$ for $\mathcal{Z}_r(t)$. With mild abuse of notation, we denote by $(\eta(t), \zeta(t))$ the random action pairs $(\alpha_\omega(t), \delta_\omega(t))$ induced by $(\eta, \zeta)$, where $(\alpha_\omega, \delta_\omega)$ is the pair of open-loop controls induced by the random seed $\omega \in \Omega_\eta \times \Omega_\zeta$ (see Lem. 2.2 of Cardaliaguet (2007) for the connection between random strategies and open-loop controls). With these, the value of the game $V : [0, T] \times \mathcal{X} \times \Delta(I) \to \mathbb{R}$ is defined by:

$$V(t_0, x_0, p) = \inf_{\{\eta_i\} \in \{\mathcal{H}_r(t_0)\}^I} \sup_{\zeta \in \mathcal{Z}_r(t_0)} \mathbb{E}_{\eta_i, \zeta} \left[ \sum_{i=1}^I p_i g_i \left( X_T^{t_0, x_0, \eta_i, \zeta} \right) + \int_{t_0}^T p_i l_i(\eta_i(s), \zeta(s)) ds \right]$$

$$= \sup_{\zeta \in \mathcal{Z}_r(t_0)} \inf_{\{\eta_i\} \in \{\mathcal{H}_r(t_0)\}^I} \mathbb{E}_{\eta_i, \zeta} \left[ \sum_{i=1}^I p_i g_i \left( X_T^{t_0, x_0, \eta_i, \zeta} \right) + \int_{t_0}^T p_i l_i(\eta_i(s), \zeta(s)) ds \right]. \tag{3}$$

**Behavioral strategy of P1.** It is proved that $V$ is a viscosity solution to a Hamilton-Jacobi Isaacs (HJI) equation with respect to $w : [0, T] \times \mathcal{X} \times \Delta(I) \to \mathbb{R}$:

$$\begin{cases} w_t + H(x, \nabla_x w) = 0 & \text{in } [0, T] \times \mathcal{X} \\ w(T, x, p) = \sum_{i=1}^{I} p_i g_i(x) & \text{in } \mathcal{X} \times \Delta(I), \end{cases} \quad (4)$$

and $V$ is convex in $p$ for all $(t_0, x_0) \in [0, T] \times \mathcal{X}$ (Cardaliaguet, 2007). The convexity of $V$ can be understood through the construction of P1's strategy. To explain, we first introduce the discrete-time Bellman backup for approximating the solution to equation 4: Let $V_\tau$ be the discrete-time approximation of $V$ with a fixed time interval $\tau$, and let

$$U_\tau(t, x, p) := \min_{u \in \mathcal{U}} \max_{v \in \mathcal{V}} \left( V_\tau(t+\tau, x+\tau f(x, u, v), p) + \tau l(u, v) \right) \quad (5)$$

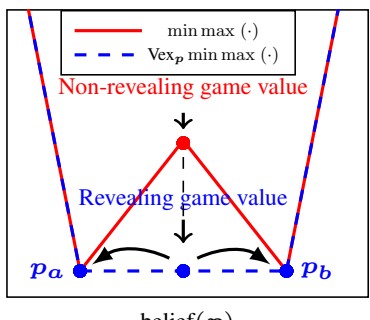

Figure 1: Revealing and non-revealing game values, and the mechanism of splitting.

be the hypothetical value if P1 plays a non-revealing strategy at $t$, i.e., P1 receives an expected payoff of $U_\tau(t, x, p)$ if he plays as if he only knows the common belief $p$ rather than the actual payoff type. Note that due to Isaacs' condition, the non-revealing game, which is complete-information, has a pure equilibrium. If $U_\tau$ is not convex in $p$, P1 can play a mixed strategy to obtain an expected payoff on the convex hull of $U_\tau$, which is an improvement from playing non-revealing. Importantly, since $p \in \Delta(I)$, the convexification can be achieved with at most $I$ vertices in $\Delta(I)$. See Fig. 1 for an illustration where $p \in \Delta(2)$. Specifically, at any $(t, x, p)$, P1 finds $\lambda = [\lambda^1, \dots, \lambda^I] \in \Delta(I)$ and splitting points $p^k \in \Delta(I)$ for $k \in [I]$ that solve

$$\min_{\lambda \in \Delta(I), \{p^k\} \in \Delta(I)^I} V_\tau(t, x, p; \lambda, \{p^k\}) := \sum_{k=1}^{I} \lambda^k \min_{u^k \in \mathcal{U}} \max_{v^k \in \mathcal{V}} \left( V_\tau(t + \tau, x^k, p^k) + \tau \mathbb{E}_{i \sim p^k}[l_i(u^k, v^k)] \right)$$

$$\text{s.t.} \quad \sum_{k=1}^{I} \lambda^k p^k = p, \quad x^k = \text{ODE}(x, \tau, u^k, v^k; f), \quad (6)$$

where $\text{ODE}(\cdot)$ computes the state at $t + \tau$. P1's strategy is to choose $u^k$ with $\Pr(u = u^k | i) = \lambda^k p^k[i] / p[i]$ if he is of type $i$.

**Remarks.** (1) If P1 announces his strategy and both players use Bayes update, one can show that belief $p$ splits to $p^k$ at $t + \tau$ if $u^k$ is chosen by P1 at $t$. (2) The relationship $V_\tau(t, x, p) = \text{Vex}_p[U_\tau(t, x, p)]$ was first introduced as the "Cav u" theorem in the context of repeated games with incomplete information (Aumann et al., 1995; De Meyer, 1996) and later extended to differential games (Cardaliaguet, 2007). However, efficient implementation of "Cav u" for differential games has not been investigated, nor does its connection with IIEFGs. While Brown et al. (2020b) discussed value convexity with respect to the public belief $p$, IIEFG solvers do not exploit "Cav u" because IIEFGs in general do not enjoy the following properties of the game we study in this paper: (i) the imperfectness of information is about the payoff types, (ii) the non-revealing games always have pure Nash, and (iii) $I \ll |\mathcal{A}|$. (3) Computing $V_\tau$ only requires pure best responses from P2 in solving $U_\tau$ at splitting points. However, these best responses cannot be considered P2's strategy.

**Behavioral strategy of P2.** For P2, the idea is to reformulate the game so that we can compute the value using P2's behavioral strategies and P1's pure best responses (at splitting points). This can be achieved by introducing the Fenchel conjugate $V^*$ of $V$:

$$V^*(t_0, x_0, \hat{p}) := \max_p p^T \hat{p} - V(t_0, x_0, p)$$

$$= \max_p p^T \hat{p} - \sup_{\zeta \in \mathcal{Z}_r(t_0)} \inf_{\{\eta_i\} \in \{\mathcal{H}_r(t_0)\}^I} \mathbb{E}_{\eta_i, \zeta} \left[ \sum_{i=1}^{I} p_i g_i \left( X_T^{t_0, x_0, \eta_i, \zeta} \right) + \int_{t_0}^{T} p_i l_i(\eta_i(s), \zeta(s)) ds \right]$$

$$= \max_p \inf_{\zeta \in \mathcal{Z}_r(t_0)} \sup_{\{\eta_i\} \in \{\mathcal{H}_r(t_0)\}^I} p^T \hat{p} - \mathbb{E}_{\eta_i, \zeta} \left[ \sum_{i=1}^{I} p_i g_i \left( X_T^{t_0, x_0, \eta_i, \zeta} \right) + \int_{t_0}^{T} p_i l_i(\eta_i(s), \zeta(s)) ds \right]$$

$$= \inf_{\zeta \in \mathcal{Z}_r(t_0)} \sup_{\eta \in \mathcal{H}(t_0)} \max_{i \in \{1, \dots, I\}} \left\{ \hat{p}_i - \mathbb{E}_{\eta, \zeta} \left[ g_i \left( X_T^{t_0, x_0, \eta, \zeta} \right) + \int_{t_0}^{T} l_i(\eta(s), \zeta(s)) ds \right] \right\}. \quad (7)$$

The last step of equation 7 uses the linearity of the payoff with respect to $p$ and also the fact that best responses are always pure (thus $\eta$ belongs to the pure strategy set $\mathcal{H}(t_0)$ rather than the random strategy set $\mathcal{H}_r(t_0)$). Critically, equation 7 describes a dual game with complete information, where the strategy space of P1 becomes $\mathcal{H}(t_0) \times [I]$, i.e., P2's goal is to minimize some worst-case dual payoff. It is proved that P2's equilibrium in the dual game starting from some $(t_0, x_0)$ is also an equilibrium for the primal game if $\hat{p} \in \partial_p V(t_0, x_0, p)$ (See App. B for explanation).

To compute P2's strategy, we first note that $V^*$ is a viscosity solution to a dual HJI equation:

$$\begin{cases} w_t + H^*(x, \nabla_x w) = 0 & \text{in } [0, T] \times \mathcal{X} \\ w(T, x, \hat{p}) = \max_i \{\hat{p}_i - g_i(x)\} & \text{in } \mathcal{X} \times \mathbb{R}^I, \end{cases} \tag{8}$$

where $H^*(x, \xi) = -H(x, -\xi)$ for any $(x, \xi) \in \mathcal{X} \times \mathbb{R}^{d_x}$. For the same reason as in the primal game, $V^*$ is convex in $\hat{p}$ for all $(t_0, x_0) \in [0, T] \times \mathcal{X}$. Similar to the primal game, at any discretized $t$, P2 finds $\lambda = [\lambda^1, \ldots, \lambda^{I+1}]$ and $\hat{p}^k \in \mathbb{R}^I$ for $k \in [I+1]$ that solve

$$\min_{\lambda \in \Delta(I+1),\ \hat{p}^k \in \mathbb{R}^I\ \forall k \in [I+1]} V^*(t, x, \hat{p}; \lambda, \{\hat{p}^k\}) := \sum_{k=1}^{I+1} \lambda^k \left( \min_{v^k \in \mathcal{V}} \max_{u^k \in \mathcal{U}} V_\tau^*(t + \tau, x^k, \hat{p}^k - \tau l(u, v)) \right)$$

$$\text{s.t.} \quad \sum_{k=1}^{I+1} \lambda^k \hat{p}^k = \hat{p}, \quad x^k = \text{ODE}(x, \tau, u^k, v^k; f), \tag{9}$$

where $l = [l_1, ..., l_I]^T$. P2's strategy is to compute the minimax solution $v^k$ corresponding to $\hat{p}^k$ and chooses $v = v^k$ with probability $\lambda^k$.

**Remarks.** (1) As we explain in App. B, the $i$th element of $\hat{p} \in \mathbb{R}^I$ represents the minimum cost P1 pays for type $i$ should P2 play her equilibrium strategy. Therefore, the dual payoff represents the $L_\infty$ distance from the primal type-dependent costs of the current strategy profile to the equilibrial costs. This suggests that $V^*(t_0, x_0, \hat{p}) = 0$ for $\hat{p} \in \partial_p V(t_0, x_0, p)$ and for any $p \in \Delta(I)$. (2) Since $\hat{p}$ is not constrained to a simplex, P2 solves a harder problem than P1. Specifically, P2 needs to find at most $I + 1$, rather than $I$, splitting points $\{\hat{p}^k\}_{k=1}^{I+1}$ in order to compute the convexification $V^*$. This is the extra cost P2 pays due to her information disadvantage. (3) Based on the interpretation of $\hat{p}$, it is clear that $\hat{p}_{t+\tau}$ splits to $\hat{p}^k - \tau l(u, v)$ with probability $\lambda^k$ when an instantaneous cost exists.

## 4 METHODS

**Reformulation of the primal and dual games.** To recap, at any $(t, x)$, P1 computes actions $u^k$ and their type-conditioned probabilities $\alpha_{ki} := \Pr(u = u^k | i)$ such that $\sum_{k=1}^I \alpha_{ki} = 1$ for $i \in [I]$. Then, $\lambda^k = \sum_{i=1}^I \alpha_{ki} p[i]$ and $p^k[i] = \alpha_{ki} p[i] / \lambda_k$ are both functions of $\alpha_{ki}$. We can now reformulate equation 6 as follows:

$$\min_{\{u^k\}, \{\alpha_{ki}\}} \max_{\{v^k\}} \sum_{k=1}^I \lambda^k \left( V(t + \tau, x^k, p^k) + \tau \mathbb{E}_{i \sim p^k}[l_i(u^k, v^k)] \right)$$

$$\text{s.t.} \quad u^k \in \mathcal{U}, \quad x^k = \text{ODE}(x, \tau, u^k, v^k; f), \quad v^k \in \mathcal{V}, \quad \alpha_{ki} \in [0, 1], \tag{P$_1$}$$

$$\sum_{k=1}^I \alpha_{ki} = 1, \quad \lambda^k = \sum_{i=1}^I \alpha_{ki} p[i], \quad p^k[i] = \frac{\alpha_{ki} p[i]}{\lambda^k}, \quad \forall i, k \in [I].$$

P$_1$ is in general a nonconvex-nonconcave minimax problem of size $(\mathcal{O}(I(I + d_u)), \mathcal{O}(I d_v))$ that needs to be solved at all sampled infostates $(t, x, p) \in [0, T] \times \mathcal{X} \times \Delta(I)$. The resultant minimax objective is by definition the convexified value of the primal game.

P2 keeps track of the dual variable $\hat{p} \in \mathbb{R}^I$ instead of the public belief $p$ during the dual game and solves the following problem at all sampled infostates $(t, x, \hat{p})$:

$$\min_{\{v^k\}, \{\lambda^k\}, \{\hat{p}^k\}} \max_{\{u^k\}} \sum_{k=1}^{I+1} \lambda^k \left( V^*(t + \tau, x^k, \hat{p}^k - \tau l(u^k, v^k)) \right)$$

$$\text{s.t.} \quad u^k \in \mathcal{U}, \quad v^k \in \mathcal{V}, \quad x^k = \text{ODE}(x, \tau, u^k, v^k; f), \quad \lambda^k \in [0, 1], \tag{P$_2$}$$

$$\sum_{k=1}^{I+1} \lambda^k \hat{p}^k = \hat{p}, \quad \sum_{k=1}^{I+1} \lambda^k = 1, \quad k \in [I + 1].$$

$P_2$ is in general nonconvex-nonconcave of size $(\mathcal{O}(I(I + d_v), \mathcal{O}(Id_u))$.

**Game solver.** Now we propose a continuous-action mixed-strategy (CAMS) solver for 2p0s differential games with one-sided information. Our algorithm performs Bellman backup through $P_1$ (resp. $P_2$) starting from the terminal condition in equation 4 (resp. equation 8) at discretized time stamps $t \in \{T, T - \tau, ..., 0\}$ and $(x, p)$ (resp. $(x, \hat{p})$) uniformly sampled in $\mathcal{X} \times \Delta(I)$ (resp. $\mathcal{X} \times \mathbb{R}^I$). Specifically, at any $t$, with a value approximation model $\hat{V}_{t+1} : \mathcal{X} \times \Delta(I) \to \mathbb{R}$, we solve $P_1$ using DS-GDA at $N$ collocation points $(x, p) \in \mathcal{X} \times \Delta(I)$ and collect a dataset $\mathcal{D}_t := \{(x^{(i)}, p^{(i)}, \tilde{V}^{(i)})\}_{i=1}^N$ where $\tilde{V}$ is the numerical approximation of the convexified value at $(t, x^{(i)}, p^{(i)})$ for the minimax problem. Then we fit a model $\hat{V}_t(x, p)$ to $\mathcal{D}_t$ and go to $t - \tau$. Alg. 1 summarizes the solver for the primal game. The dual game solver is similarly defined. We note that our algorithm does not search along the game tree as in (Schmid et al., 2023; Brown et al., 2020b). This is because value approximation over the entire $[0, T] \times \mathcal{X} \times \Delta(I)$ is necessary for long-term safety planning in risk-sensitive applications, e.g., when state constraints are imposed. We will introduce a state-constrained variant of the game in Sec. 6.

---

**Algorithm 1:** Continuous Action Mixed Strategy solver (CAMS)

---

**Input:** time discretization $\tau$, terminal value $V(T, \cdot, \cdot)$, sample size $N$, minimax solver $\mathbb{O}$
**Initialize:** value network $\{\hat{V}_t\}_{t=0}^{T-\tau}$, training dataset $\mathcal{D} \leftarrow \emptyset$
$\mathcal{S} \leftarrow$ sample $N$ states $(x, p) \in \mathcal{X} \times \Delta(I)$;
**for** $t$ *in* $\{T - \tau, ..., 0\}$ **do**
    **for** $(x, p)$ *in* $\mathcal{S}$ **do**
        $\vartheta \leftarrow \mathbb{O}(t, x, p)$ ;                         /* Solution to P$_1$ */
        append $\{(t, x, p), \vartheta\}$ to $\mathcal{D}$;
    Fit $\hat{V}_t$ to $\mathcal{D}$

---

## 5 PREDICTION ERROR AND COMPLEXITY OF CAMS

Here we show that CAMS shares the same exponential error propagation as in standard approximate value iteration (AVI). The only difference is that the measurement error in CAMS comes from numerical approximation of the minimax problems rather than randomness in state transition and rewards. To start, let the true value be $V(t, x, p)$. Following Zanette et al. (2019), the prediction error $\epsilon_t^{bias} := \max_{x,p} |\hat{V}_t(x, p) - V(t, x, p)|$ is affected by (1) the prediction error $\epsilon_{t+\tau}^{bias}$ propagated back from $t + \tau$, (2) the minimax error $\epsilon_t^{minmax}$ caused by limited iterations in solving the minimax problem at each collocation point: $\epsilon_t^{minmax} = \max_{(x,p) \in \mathcal{D}_t} |\tilde{V}(t, x, p) - V(t, x, p)|$, and (3) the approximation error due to the fact that $V(t, \cdot, \cdot)$ may not lie in the model hypothesis space $\mathcal{V}_t$ of $\hat{V}_t$: $\epsilon_t^{app} = \max_{x,p} \min_{\hat{V}_t \in \mathcal{V}_t} |\hat{V}_t(x, p) - V(t, x, p)|$.

**Approximation error.** For simplicity, we will abuse the notation by using $x$ in place of $(x, p)$ and omit time dependence of variables when possible. In practice we consider $\hat{V}_t$ as neural networks that share the architecture and the hypothesis space. Note that $\hat{V}_T(\cdot) = V(T, \cdot)$ is analytically defined by the boundary condition in equation 4 and thus $\epsilon_T^{app} = \epsilon_T^{bias} = 0$. To enable the analysis on neural networks, we adopt the assumption that $\hat{V}$ is infinitely wide and that the resultant neural tangent kernel (NTK) is positive definite. Therefore from NTK analysis (Jacot et al., 2018), $\hat{V}$ can be considered a kernel machine equipped with a kernel function $r(x^{(i)}, x^{(j)}) := < \phi(x^{(i)}), \phi(x^{(j)}) >$ defined by a feature vector $\phi : \mathcal{X} \to \mathbb{R}^{d_\phi}$. Given training data $\mathcal{D} = \{(x^{(i)}, V^{(i)})\}$, let $r(x)[i] := r(x^{(i)}, x^{(j)})$, $R_{ij} := r(x^{(i)}, x^{(j)})$, $V_{\mathcal{D}} := [V^{(1)}, ..., V^{(N)}]^\top$, $\Phi_{\mathcal{D}} := [\phi(x^{(1)}), ..., \phi(x^{(N)})]$, and $w_{\mathcal{D}} := \Phi_{\mathcal{D}}(\Phi_{\mathcal{D}}^\top \Phi_{\mathcal{D}})^{-1} V_{\mathcal{D}}$ be model parameters learned from $\mathcal{D}$, then

$$\hat{V}(x) = r(x)^\top R^{-1} V_{\mathcal{D}} = < \phi(x), w_{\mathcal{D}} > \tag{10}$$

is a linear model in the feature space. Let $\theta^{\phi(x)} := r(x)^\top R^{-1}$ and $C := \max_x \|\theta^{\phi(x)}\|_1$. Further, let $\mathcal{D}^* := \arg\min_{\mathcal{D}} | < \phi(x), w_{\mathcal{D}} > -V(x)|$ and $w^* := w_{\mathcal{D}^*}$, i.e., $w^*$ represents the best hypothetical model given sample size $N$. Since $N$ is finite, the data-dependent hypothesis space induces an approximation error $\epsilon_t^{app} := \max_x | < \phi(x), w^* > -V(x)|$.

**Error propagation.** Recall that we approximately solve $P_1$ at each collocation point. Let $z := \{\lambda, p, u, v\}$ be the collection of variables and $\tilde{z}$ be the approximated saddle point resulted from DS-GDA. Let $\tilde{V}(t, x, \tilde{z})$ be the value approximated at $(t, x)$ and $V(t, x, z^*)$ be the value at the true saddle point $z^*$. Lem. 1 bounds the error of $\tilde{V}(t, x, \tilde{z})$:

**Lemma 1.** $\max_x |\tilde{V}(t, x, \tilde{z}) - V(t, x, z^*)| \leq \epsilon_{t+\tau}^{bias} + \epsilon_t^{minmax}$.

Now we can combine this measurement error with the inherent approximation error $\epsilon_t^{app}$ to reach the following bound on the prediction error $\epsilon_t^{bias}$:

**Lemma 2.** $\max_x |\hat{V}_t(x) - V(t, x)| \leq C_t(\epsilon_t^{minmax} + \epsilon_{t+\tau}^{bias} + \epsilon_t^{app}) + \epsilon_t^{app}$.

Lem. 3 characterizes the propagation of error:

**Lemma 3.** Let $\epsilon_t^{app} \leq \epsilon^{app}$, $\epsilon_t^{minmax} \leq \epsilon^{minmax}$, and $C_t \leq C$ for all $t \in [T]$. If $\epsilon_T^{app} = 0$, then $\epsilon_0^{bias} \leq TC^T(\epsilon^{app} + C(\epsilon^{minmax} + \epsilon^{app}))$.

We can now characterize the computational complexity of CAMS through Thm. 1, by taking into account the number of DS-GDA iterations and the per-iteration complexity:

**Theorem 1.** *With a computational complexity of $\mathcal{O}(TNI^2\epsilon^{-4})$, CAMS achieves*

$$\max_{(x,p)\in\mathcal{X}\times\Delta(I)} |\hat{V}_0(x, p) - V(0, x, p)| \leq TC^T(\epsilon^{app} + C(\epsilon^{app} + \epsilon)). \tag{11}$$

**Remarks.** Proofs all deferred to App. D. From Thm. 1, value prediction error grows exponentially when $C > 1$. Zanette et al. (2019) discussed a linear value approximator that achieves $C = 1$. However, their method requires solving a linear program (LP) for every inference $\hat{V}_t(x, p)$ if $(x, p)$ does not belong to the training set. For CAMS, incorporating this method would require auto-diff through the LP solver during each descent and ascent steps in solving the minimax problems, which turned out to be expensive in PyTorch and JAX. While effectively suppressing $C$ for neural nets remains to be investigated, the superiority of CAMS against SOTA IIEFG solvers can already be demonstrated when action spaces are continuous, as we discuss in Sec. 6.

## 6 EMPIRICAL VALIDATION

We introduce Hexner's game (Hexner, 1979) that has an analytical Nash equilibrium. We use variants of this game to compare CAMS with baselines (MMD, CFR+, and DeepCFR) on solution quality and computational cost. We then demonstrate the scalability of CAMS using a high-dimensional version of the game.

### 6.1 HEXNER'S GAME

Hexner (1979) introduced a homing problem where the system dynamics is decomposed as $\dot{x}_j = A_j x_j + B_j u_j$ for $j = [2]$, where $x_j \in \mathcal{X}_j$, $u_j \in \mathcal{U}_j$, and $A_j$ and $B_j$ are known matrices. P1's target states is $z\theta$ where $\theta$ is drawn with distribution $p_0$ from $\Theta$, $|\Theta| = I$, and $z \in \mathbb{R}^{d_x}$ is fixed and common knowledge. Denote by $\eta_i(t)$ and $\zeta(t)$ the random actions at time $t$ induced by strategy pair $(\eta_i, \zeta)$. The expected payoff to P1 is:

$$J(\{\eta_i\}, \zeta) = \mathbb{E}_{i\sim p_0}\left[ \int_0^T (\eta_i(t)^\top R_1 \eta_i(t) - \zeta(t)^\top R_2 \zeta(t))dt + [x_1(T) - z\theta_i]^\top K_1(T) [x_1(T) - z\theta_i] \right.$$

$$\left. - [x_2(T) - z\theta_i]^\top K_2(T) [x_2(T) - z\theta_i] \right],$$

$$\tag{12}$$

where $R_1, R_2 \succ 0$ are control-penalty matrices and $K_1, K_2 \succeq 0$ are state-penalty matrices. Essentially, the goal of P1 is to get closer to the target $z\theta$ than P2. To take full information advantage, P1 needs to decide when to home-in to and thus reveal the target. See Fig. 2 for an illustration. As explained in Hexner (1979); Ghimire et al. (2024), this game has an analytical solution: Given a problem-dependent critical time $t_r := t_r(T, \{A_j\}, \{B_j\}, \{R_j\}, \{K_j\})$, if $t_r \in (0, T)$, P1 homes towards the mean target $\mathbb{E}[\theta]$ as if he does not know the actual target until $t_r$. Otherwise, P1 homes towards the actual target immediately at $t = 0$. P2's strategy is to follow P1.

### 6.2 COMPARISONS ON 1- AND 4-STAGE HEXNER'S GAMES

**Settings.** We first use a normal-form Hexner's game with $\tau = T$ and a fixed initial state $x_0 \in \mathcal{X}$ to demonstrate that IIEFG algorithms suffer from increasing costs along $|\mathcal{A}|$ while CAMS does not. The baselines we consider include CFR+ (Tammelin, 2014), MMD (Sokota et al., 2022), and a modified CFR-BR (Johanson et al., 2012) (dubbed CFR-BR-Primal), where we only compute P2's best response to P1's current strategy and only focus on converging P1's strategy, which matches with CAMS for solving $P_1$. All baselines are implemented in OpenSpiel (Lanctot et al., 2019). The normal-form primal game has a trivial ground-truth strategy where P1 goes directly to his target. For visualization, we use $d_x = 4$ (position and velocity in 2D). For baselines, we use discrete action sets $\mathcal{A}_1 = \mathcal{A}_2$ defined by lattices on $\mathcal{U}_1 \times \mathcal{U}_2$ with 4 lat-

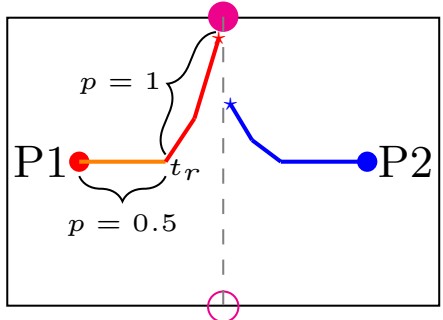

Figure 2: Sample equilibrium in Hexner's game. Magenta circles are target states with $p_0 = [0.5, 0.5]^\top$. Public belief is kept $p_0$ until $t_r$ and becomes $[1, 0]^\top$ when P1 starts to move to the true target.

tice sizes so that $|\mathcal{A}_j| = \{16, 36, 64, 144\}$. All algorithms terminate when a threshold of NashConv is met. For conciseness, we only consider solving P1's strategy and thus use P1's $\delta$ in NashConv. We set the threshold to $10^{-3}$ for baselines and $10^{-5}$ for CAMS. We will show that even with a more stringent threshold, CAMS still converges significant faster than the baselines. We then use Deep-CFR as a baseline for a Hexner's game with 4 time steps, where $T = 1$ and $\tau = 0.25$. DeepCFR were run for 1000 CFR iterations (resp. 100) with 10 (resp. 5) traversals for $|\mathcal{A}_j| = 9$ (resp. 16). More details on experiment settings can be found in App. E.3.

**Comparison metrics.** For the normal-form game, we compare both computational cost and the expected action error $\varepsilon$ from the ground-truth action of P1: $\varepsilon(x_0) := \mathbb{E}_{i \sim p_0} \left[ \sum_{k=1}^{|\mathcal{A}|} \alpha_{ki} \|u_k - u_i^*(x_0)\|_2 \right]$, where $u_i^*(x_0)$ is the ground truth for type $i$ at $x_0$. For the 4-stage game, we compare the expected action errors at each time step: $\bar{\varepsilon}_t := \mathbb{E}_{x_t \sim \pi}[\varepsilon(x_t)]$, where $\pi$ is the strategy learned by DeepCFR or CAMS. For each strategy, we estimate $\{\bar{\varepsilon}_t\}_{t=1}^4$ by generating 100 trajectories with initial states uniformly sampled from $\mathcal{X}$. The wall-time costs for game solving are 17 hours using CAMS, 29 hours ($|\mathcal{A}| = 9$) and 34 hours ($|\mathcal{A}| = 16$) using DeepCFR, all on an A100 GPU.

**Results.** Fig. 3 summarizes the comparisons. For the normal-form game, all baselines have complexity and wall-time costs increasing with $|\mathcal{A}|$, while CAMS is invariant to $|\mathcal{A}|$. With the similar or less compute, CAMS achieves significantly better strategies than DeepCFR in the 4-stage game. Fig. 4 visualizes sample trajectories for the 4-stage game.

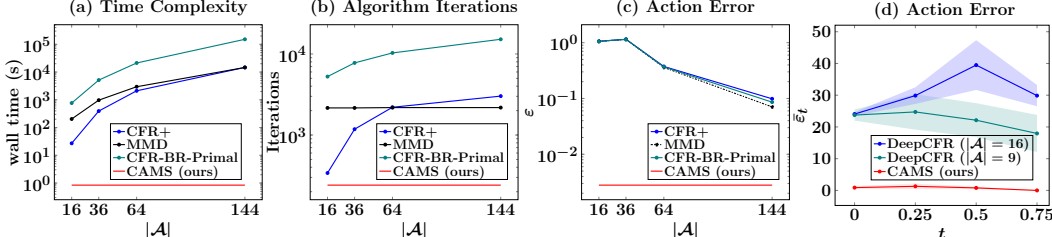

Figure 3: (a-c) Comparisons b/w CAMS (ours), CFR+, MMD, CFR-BR-Primal on 1-step Hexner's game. (d) Comparison b/w CAMS and DeepCFR on 4-stage Hexner's w/ similar compute.

### 6.3 SCALABILITY OF CAMS ON 10-STAGE AND 3D VARIANTS OF HEXNER'S GAME

**10-stage game.** Here we solve Hexner's games with $T = 1$ and $\tau = 0.1$, and consider both state-constrained and -unconstrained cases. These games have a game-tree complexity of $10^{80}$ if we use an action discretization of $|\mathcal{A}_j| = 10k$ (100 discrete values along each of the two action dimensions). In the state-constrained version of the game, P1 receives $+\infty$ if he collides with P2. Collision occurs when the Euclidean distance between the players is less than 0.05. As a result, the Nash equilibrium of this game variant is no longer analytical. Following Ghimire et al. (2024), we approximate a time-dependent safe zone $\Omega_t \subseteq \mathcal{X}$ for P1 so that for any initial state outside of $\Omega_t$, P1 surrenders because for any P1's strategy, P2 can always find a strategy to collide. Within $\Omega_t$, the Nash equilibrium can

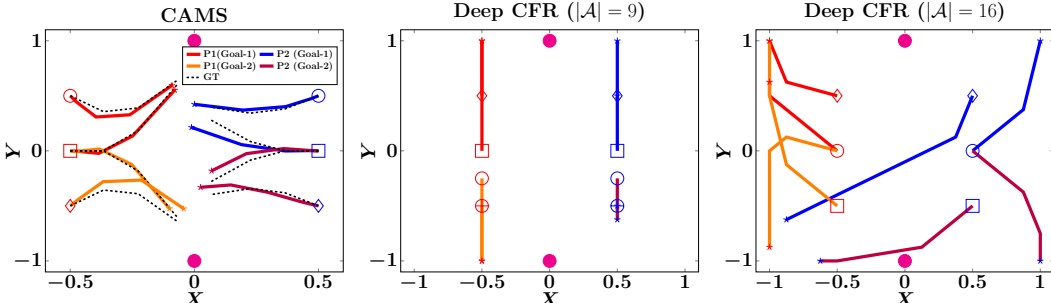

Figure 4: Trajectories generated using CAMS (primal game) and DeepCFR. The initial position pairs are marked with same marker and the final with star. The trajectories from CAMS are close to the ground-truth while DeepCFR fails to converge with even more compute.

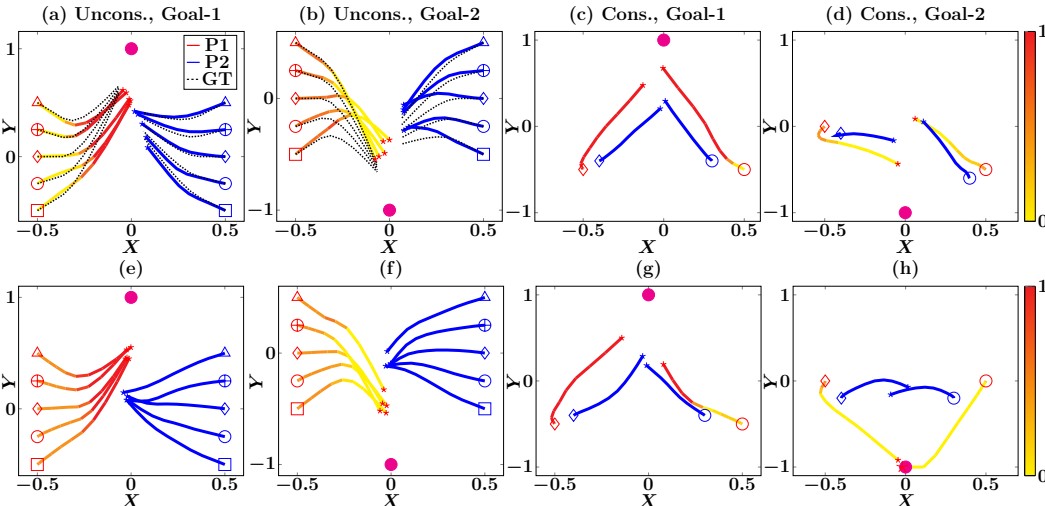

Figure 5: Sample trajectories for the primal game (a-d) where P1 plays Nash and P2 plays best response, and primal-dual game (e-h) where both players play Nash. Cols 1 & 2 are unconstrained, cols 3 & 4 are w/ collision constraint. Dotted lines are ground-truth Nash. Color shades indicate evolution of public belief (1 means Goal-1). Initial position pairs are marked with same markers.

be derived from CAMS where for each minimax problem, P1's admissible actions are restricted by $\Omega_t$. In Ghimire et al. (2024), the resultant constrained minimax problems are solved as follows: First, at each $t$, non-revealing games (without splitting) are approximately solved across $\mathcal{X} \times \Delta(I)$ using an enumeration over $|\mathcal{A}_j| = 100$ for $j \in [2]$. This resulted in finding the minimax point from a $100 \times 100$ matrix for each $(x, p)$. Then with the resultant values for the non-revealing games, the value convex hulls over the public belief is approximated for each sampled $x$, before $\hat{V}_t$ is fit to these approximated convex hulls. Due to the use of enumeration, this method has exponential space and computational complexities with respect to the dimensionalities of the action spaces. In this paper, we solve $P_1$ (and $P_2$) which directly approximates the convexified values, avoiding errors introduced by numerical convexification. In addition, since DS-GDA is gradient-based, the resultant space and computational complexities are only linear to the dimensionality of the action spaces. **Results:** Results are summarized in Fig. 5. For the unconstrained game where analytical strategies are known, we compare the approximated and the ground-truth strategies starting from various initial states. While approximation errors exist, CAMS successfully learns the target-concealing behavior of P1 as P1 always moves towards $\mathbb{E}_{p_0}[z\theta]$ before revealing his target. Averaging over 50 trajectories derived from CAMS, P1 conceals the target until $t_r = 0.60s \pm 0.06s$ (compared to the ground-truth $t_r = 0.5s$). CAMS also approximates P2's robust strategy well, as P2 only starts to home towards a target after P1 reveals. We note that the complexity of the dual game is higher than that of the primal

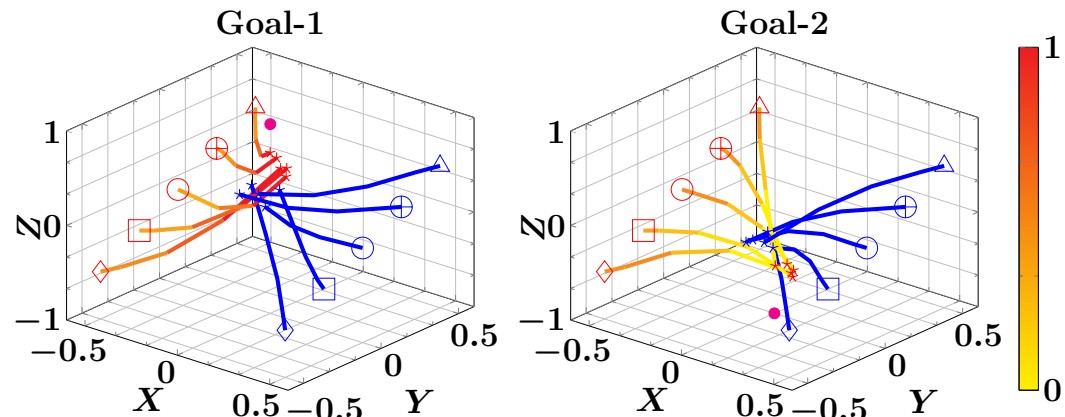

Figure 6: 3D Hexner's Game. Color shades indicate the current public belief.

game because its value is one dimension higher and $P_2$ is larger than $P_1$. This resulted in higher error in approximating P2's strategies.

**3D Hexner's game.** To demonstrate the scalability of CAMS, we solve a 3D Hexner's game where the joint action space is now 6D. Accordingly, the state space becomes 12D and the value becomes 13D. Resultant trajectories are visualized in Fig. 6. Similar to the 2D case, P1 learns to correctly conceal his target until some critical time.

## 7 CONCLUSION

Unlike IIEFG where mixed strategies have to be approximated over the entire action space across the game tree, we showed that differential games with one-sided information on game type enjoy a much simpler strategy structure when the Isaacs' condition holds: The strategy of the informed (resp. uninformed) player has at most $I$ (resp. $I+1$) pure action branches at each infostate. We demonstrated the clear advantage of using this structural property in solving games with continuous action spaces, against SOTA IIEFG solvers, in terms of computational cost and solution quality. To the authors' best knowledge, this is the first method that enables tractable solve of incomplete-information games with continuous action spaces without problem-specific abstraction and discretization.

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

## A  CONNECTION BETWEEN VALUE CONVEXIFICATION AND NASH EQUILIBRIUM IN INCOMPLETE-INFORMATION GAMES

Here we explain the construction of Nash equilibrium as a consequence of value convexification. For ease of exposition, we will use examples from a simplistic setting: repeated normal-form games with one-sided information. We also walk through the computation of strategies for the informed and uninformed players for the given examples. We refer readers to Aumann et al. (1995); De Meyer (1996); Sorin (2002) for more details on the theoretical development.

Consider two normal-form zero-sum payoff tables given by matrices $G_1$ and $G_2$ as shown in equation 13. P1 is the row player with actions $\{U, D\}$ and P2 the column player with actions $\{L, R\}$. At the beginning of the game, nature picks game $G_1$ with probability $p$ and communicates that only to P1. P2 only knows the probability $p$. Both players pick their actions and announce them simultaneously for that round without knowing the resultant payoff. This process is repeated until the end of the game, at which point the average payoff is revealed. The game can be repeated either finitely or infinitely. For conciseness, we only discuss the latter case. To align the discussion with literature on repeated games, we will consider P1 maximize, rather than minimize, the payoff. We call this game $G(p)$.

$$G_1 = \begin{array}{c} \\ A \\ B \end{array} \begin{array}{cc} L & R \\ \left[\begin{array}{cc} 1 & 0 \\ 0 & 0 \end{array}\right] \end{array} \qquad G_2 = \begin{array}{c} \\ A \\ B \end{array} \begin{array}{cc} L & R \\ \left[\begin{array}{cc} 0 & 0 \\ 0 & 1 \end{array}\right] \end{array} \tag{13}$$

Let us assume, for simplicity, $p = 0.5$, and that the game being played is $G_1$. Since P1 knows that $G_1$ is the game, he could play $A$ every time, as $B$ would otherwise lead to a payoff of zero. However, as the game progresses, P2 will be able to deduce that $G_1$ is the game being played, forcing her to always play $R$, which guarantees a payoff of zero. Similarly, if $G_2$ is selected, and P1 always plays $B$, P2 will eventually figure out the true game, and guarantee a payoff of zero in the remainder of the game. In this particular game, P1 can improve his expected payoff by ignoring the actual game type. Then players play a complete-information game given by the expected payoff matrix $\bar{G}(p) = \begin{bmatrix} \frac{1}{2} & 0 \\ 0 & \frac{1}{2} \end{bmatrix}$, for which the optimal strategy for P1 (resp. P2) is to play $\{A, B\}$ (resp. $\{L, R\}$) with probability 0.5, leading to an expected payoff of $\frac{1}{4}$ to P1 in each round. Notice that by playing this way, P1 conceals the information about which game is being played, i.e., the public belief $p$ is always 0.5. Thus $\frac{1}{4}$ is the value of the *non-revealing* game and the corresponding strategy is known as the *non-revealing* strategy. In the above game, the non-revealing strategy is Nash. One can easily see that in the game $(-G_1, -G_2)$, a revealing strategy of P1 will instead be Nash.

It is important to note that for some games P1 will partially reveal the type information by splitting the belief in the first round. This can be seen from the following game with two possible payoff tables in equation 14:

$$G_1 = P1 \begin{array}{c} P2 \\ \begin{bmatrix} 0 & 1 & 1 & 3 \\ 0 & 1 & 0 & 3 \end{bmatrix} \end{array} \qquad G_2 = P1 \begin{array}{c} P2 \\ \begin{bmatrix} 3 & 0 & 1 & 0 \\ 3 & 1 & 1 & 0 \end{bmatrix} \end{array} \tag{14}$$

Let $p$ be the probability that the chosen game is $G_1$. Then the non-revealing game is defined by:

$$\bar{G}(p) = P1 \begin{array}{c} P2 \\ \begin{bmatrix} 3(1-p) & p & 1 & 3p \\ 3(1-p) & 1 & (1-p) & 3p \end{bmatrix} \end{array}. \tag{15}$$

Let $U(p)$ be the value of the non-revealing game, and let $V(p)$ be the value of the original game. Theorem 3.2 in Aumann et al. (1995) says that $V(p)$ is the concave hull of $U(p)$, i.e., for any $p \in \Delta(I)$ ($I = 2$ in this case)

$$V(p) = \text{Cav} \, U(p). \tag{16}$$

This is because for any $p \in \Delta(I)$ where $U(p) < \text{Cav} \, U(p)$, P1 can play a mixed strategy to achieve an expected payoff of $\text{Cav} \, U(p)$, by splitting the public belief to some $I$ vertices in $\Delta(I)$. Once this splitting is done, P1 can keep on playing non-revealing strategy to maintain $\text{Cav} \, U(p)$ as his expected payoff. We elaborate using the example: The value of the non-revealing game $U(p)$ is

$$U(p) = \begin{cases} 3p, & 0 \le p \le 2 - \sqrt{3} \\ 1 - p(1-p), & 2 - \sqrt{3} \le p \le \sqrt{3} - 1 \\ 3(1-p), & \sqrt{3} - 1 \le p \le 1. \end{cases} \tag{17}$$

The concavification of the value is given by:

$$\text{Cav} \, U(p) = \begin{cases} 3p, & 0 \le p \le 2 - \sqrt{3} \\ 6 - 3\sqrt{3}, & 2 - \sqrt{3} \le p \le \sqrt{3} - 1 \\ 3(1-p), & \sqrt{3} - 1 \le p \le 1. \end{cases} \tag{18}$$

Both $U(p)$ and $V(p)$ are visualized in Fig 7. From the figure, P1 attains maximum value $6 - 3\sqrt{3}$ at $p = 2 - \sqrt{3}$ and $p = \sqrt{3} - 1$. Therefore, P1 can play a mixed strategy to attain the maximum value by announcing a mixed strategy in such a way that the public belief $p$ is updated to either $(2 - \sqrt{3})$ or $(\sqrt{3} - 1)$ depending on the action P1 actually takes. This makes P1's strategy *partially revealing* as P2 will not be able to deduce P1's true type. Specifically, for $p = 0.5$, and if the

actual game is $G_1$, P1 plays the mixed strategy for $\bar{G}(2 - \sqrt{3})$ with the probability $2 - \sqrt{3}$ and for $\bar{G}(\sqrt{3} - 1)$ with probability $\sqrt{3} - 1$; if the actual game is $G_2$, he plays the mixed strategy for $\bar{G}(2 - \sqrt{3})$ with probability $\sqrt{3} - 1$ and for $\bar{G}(\sqrt{3} - 1)$ with probability $2 - \sqrt{3}$. More generally, for any nature's distribution $p$, P1's strategy is to compute $\lambda \in \Delta(I)$ and $p_i \in \Delta(I)$ such that $\sum_{i=1}^{I} \lambda[i] u(p^i) = \text{Cav}(U(p))$ and $\sum_{i=1}^{2} \lambda_i p^i = p$. Then, given his true type $k$, he plays the maximin strategy for $\bar{G}(p^i)$ with probability $\lambda_i p_k^i / p_k$. Gilpin & Sandholm (2008) first discussed the nonconvex problem for solving Cav $u$.

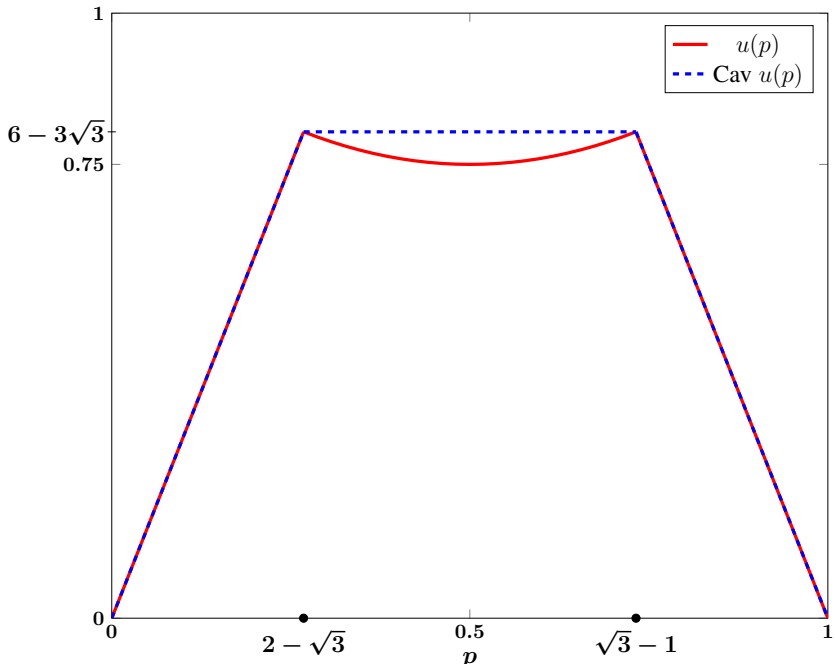

Figure 7: Non-revealing game value $u_1$ and its concavification

Next, we need to derive strategy for P2. Unlike P1, P2 has to guess the true game that is being played and hedge against potential manipulation from P1. A good strategy is to play in such a way that she pays the same amount to P1 no matter the type of the game. To do so, P2 plays a game with a vector payoff that contains the amount she pays to P1 for each game types.

Consider the game in equation 14. By observing P1's action, P2 can keep track of the vector payoffs $(x, y)$ for each stage. If at the beginning of the game P1 chose the last row and P2 chose the last column, then the vector payoff is $(3, 0)$. All possible vector payoffs define vertices in Fig. 8. The running average of the vector payoffs (the shaded region in Fig. 8) is defined by:

$$(\xi_n, \eta_n) = \left( \frac{1}{n}(x_1 + x_2 + \cdots + x_n), \frac{1}{n}(y_1 + y_2 + \cdots + y_n) \right).$$

P2 knows that if $G_1$ (resp. $G_2$) is the game, P1 will move the average to the right (resp. top). Blackwell (1956) first discussed P2's strategy to minimize the average payoff by introducing the concept of *approachability*: A set $S$ in the payoff vector space is *approachable* for P2 if P2 can adopt a strategy ensuring that the distance of the running vector payoff from $S$ converges to zero with probability one, regardless of P1's strategy.

From the primal game, we know that P1 can guarantee payoff of $6 - 3\sqrt{3}$ (the dashed lines in Fig. 9). To construct the approachable set of P2, consider P1's mixed strategy as $(\pi, 1 - \pi)$ and P2's mixed strategy as $(\sigma_1, \sigma_2, \sigma_3, \sigma_4)$. We can determine the expected payoffs to P1: When P2 plays first column, the payoff to P1 is $(0, 3)$, when she plays the second, it is $(1, 1 - \pi)$, and so on. Thus, for all possible $(\sigma_1, \sigma_2, \sigma_3, \sigma_4)$, the expected payoffs to P1 is the convex hull of the points $(0, 3), (1, 1 - \pi), (\pi, 1 - \pi), (3, 0)$. Denote the shaded region in Fig. 9 as $S = \{\xi_n, \eta_n : (\xi_n, \eta_n) \leq 6 - \sqrt{3}\}$.

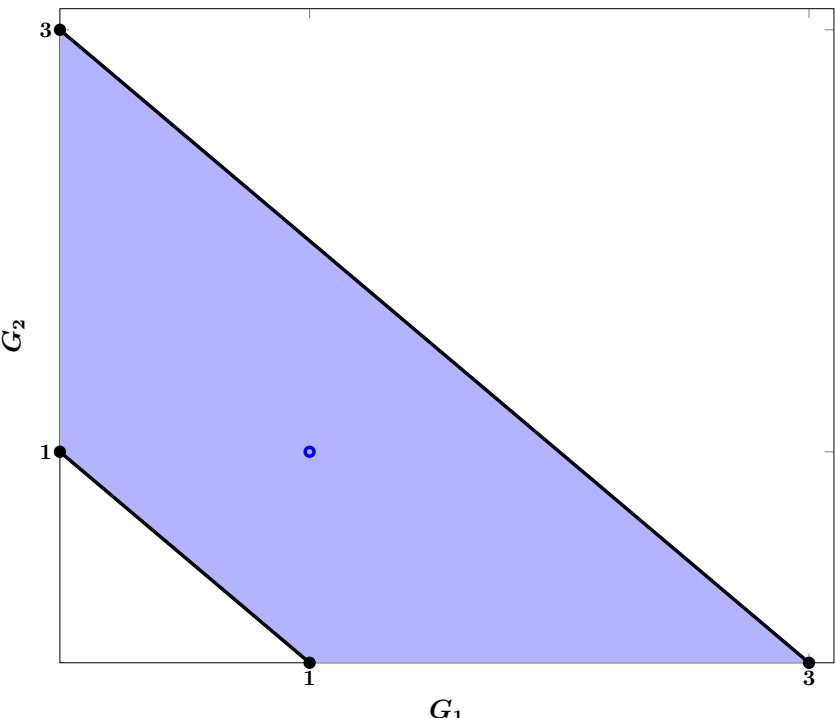

Figure 8: Game from P2's perspective.

The optimal strategy for P2 is as follows. P2 keeps track of average vector payoff (say $g_n = (\xi_n, \eta_n)$). If $g_n \in S$, then P2 plays arbitrarily. However, if $g_n \notin S$, P2 must project the vector $g_n$ onto the closest point $c = \arg\min_{m \in C} ||g_n - m||$. P2 then adopts the mixed strategy corresponding to the projection $q = (g_n - c)/||g_n - c|| \in \Delta(K)$ (here, $K = 2$), and plays optimally in the game $G(q)$.

## B CONNECTION BETWEEN PRIMAL AND DUAL GAMES

Here we continue to use the infinitely-repeated game setting to explain the connection between the primal and the dual games and the interpretation of the dual variable $\hat{p}$. Please see Theorem 2.2 in De Meyer (1996) and the extension to differential games in Cardaliaguet (2007).

Let the primal game be $G(p)$ for $p \in \Delta(I)$, the dual game be $G^*(\hat{p})$ for $\hat{p} \in \mathbb{R}^I$, and let $\{\eta_i\}_{i=1}^I$ be the set of strategies for P1 and $\zeta$ the strategy for P2. $\eta_i \in \Delta(d_u)$ and $\zeta \in \Delta(d_v)$. We note that P1's strategy $\{\eta_i\}_{i=1}^I$ can also be together represented in terms of $\pi := \{\pi_{ij}\}^{I,d_u}$ such that $\sum_j^{d_u} \pi_{ij} = p[i]$ and $\eta_i[j] = \pi_{ij}/p[i]$, i.e., nature's distribution is the marginal of $\pi$ and P1's strategy the conditional of $\pi$. Let $G_{\eta\zeta}^i$ be the payoff to P1 of type $i$ for strategy profile $(\eta, \zeta)$. We have the following results connecting $G(p)$ and $G^*(\hat{p})$:

1. If $\pi$ is Nash for P1 in $G(p)$ and $\hat{p} \in \partial V(p)$, then $\{\eta_i\}_{i=1}^I$ is also Nash for P1 in $G^*(\hat{p})$.

2. If $\pi$ is Nash for P1 in $G^*(\hat{p})$ and $p$ is induced by $\pi$, then $p \in \partial V^*(\hat{p})$ and $\pi$ is Nash for P1 in $G(p)$.

3. If $\zeta$ is Nash for P2 in $G^*(\hat{p})$ and $p \in \partial V^*(\hat{p})$, then $\zeta$ is also Nash for P2 in $G(p)$.

4. If $\zeta$ is Nash for $G(p)$, and let $\hat{p}^i := \max_{\eta \in \Delta(d_u)} G_{\eta\zeta}^i$ and $\hat{p} := [\hat{p}^1, ..., \hat{p}^I]^T$, then $p \in \partial V^*(\hat{p})$ and $\zeta$ is also Nash for P2 in $G^*(\hat{p})$.

From the last two properties we have: If $\zeta$ is Nash for $G(p)$ and $G^*(\hat{p})$, then $\hat{p} = \max_{\eta \in \Delta(d_u)} G_{\eta\zeta}^i$, i.e., $\hat{p}[i]$ is the payoff of type $i$ if P1 plays a best response for that type to P2's Nash.

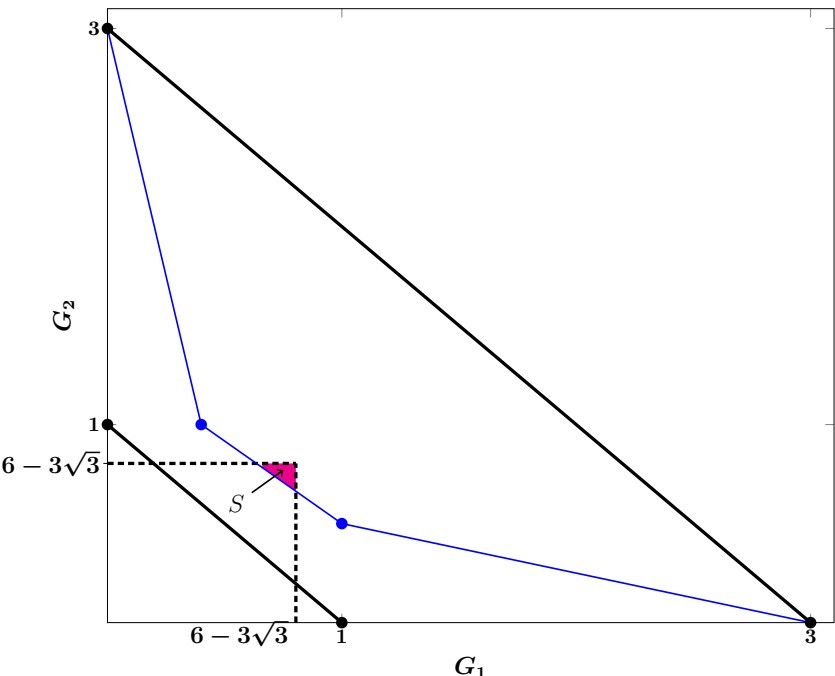

Figure 9: Approachable set (shaded in magenta) of P2

## C  COMPUTATIONAL COMPLEXITY OF EXISTING ALGORITHMS FOR SOLVING 2P0S NORMAL FORM GAMES

Here we reveal the computational complexity (in terms of the number of iterations) of some important existing algorithms for solving 2p0s normal form games. The purpose is to show that these algorithms all scale with the action space size, which limits them from solving games with continuous action spaces with discretization leads to undesirable solutions. We omit discussions about IIEFGs since they can be reformulated as NFGs.

Consider the following minimax formulation for NFGs:

$$\min_{x \in \Delta(I)} \max_{y \in \Delta(J)} x^T A y + \alpha g_1(x) - \alpha g_2(y), \tag{19}$$

where $I$ and $J$ are positive integers, $A \in \mathbb{R}^{I \times J}$ is a payoff matrix, and $g_1$, $g_2$ are strictly convex functions (e.g., L2 norm, negative entropy for NFGs, and dilated entropy for EFGs). Since equation 19 is convex to $x$ and concave to $y$, there exists a unique solution. When $\alpha = 0$, the solution $(x^*, y^*)$ is a Nash equilibrium, otherwise if $\alpha > 0$, the solution is an quantal response equilibrium (QRE).

**Counterfactual regret minimization**  CFR variants are average-time convergent algorithms for solving NFGs and EFGs, leveraging the fact that minimizing counterfactual regrets at all infostates achieves Nash for 2p0s games (Zinkevich et al., 2007). **Algorithm:** Here we introduce the standard CFR and CFR+. For simplicity, we will focus on solving the NFG in equation 19 with $\alpha = 0$ (which reduces CFR to regret matching and CFR+ to regret matching+). Given strategy profile $(x_t, y_t)$ at iteration $t \in [T]$, the instantaneous regret vector for Player 1 (resp. Player 2) is $r_1^t = A y_t - x_t^T A y_t$ (resp. $r_2^t = A^T x_t - x_t^T A y_t$). The non-negative regret vector is $R_i^t = \max\{\sum_{\tau=1}^{t} r_i^\tau, 0\}$ for $i \in [2]$. CFR updates the strategies as

$$x_{t+1} = \frac{R_1^t}{<\mathbf{1}, R_1^t>}, \quad y_{t+1} = \frac{R_2^t}{<\mathbf{1}, R_2^t>} \tag{20}$$

if the sums $<\mathbf{1}, R_i^t>$ is positive. Otherwise the strategy is updated as $x_{t+1} = \mathbf{1}/I$ for Player 1 and $y_{t+1} = \mathbf{1}/J$ for Player 2. CFR+ is different from CFR only in the definition of the instantaneous

regret: $\hat{r}_i^t = \max\{r_i^t, 0\}$ and then $R_i^t = \max\{\sum_{\tau=1}^t \hat{r}_i^\tau, 0\}$. **Complexity:** To reach $\varepsilon$-Nash, the best-known upper bound on the complexity of CFR and CFR+ is $\mathcal{O}((I+J)/\varepsilon^2)$ (Cesa-Bianchi & Lugosi, 2006). While this sublinear convergence rate seems to be worse than regularized descent-ascent algorithms with guaranteed linear convergence (e.g., MMD and regularized FTRLs), CFR+ still enjoys the state-of-the-art empirical performance for a variety of large IIEFGs (Tammelin, 2014). Nonetheless, it should be noted that the complexity of CFR variants scales linearly with respect to the size of the action space.

**Magnetic mirror descent** MMD is an extension of projected gradient descent ascent that has linear last-iterate convergence to $\alpha$-QRE for $\alpha > 0$. For ease of exposition, we set $g_1(x) = \frac{1}{2}\|x\|_2^2$ and $g_2$ is similarly defined [1] **Algorithm:** Let $\eta > 0$ be a learning rate, $(x', y') \in \text{int } \Delta(I) \times \Delta(J)$ be a "magnet". Then starting from $(x_1, y_1) \in \text{int } \Delta(I) \times \Delta(J)$, at each iteration $t \in [T]$ do

$$
\begin{aligned}
x_{t+1} &= \arg\min_{x \in \Delta(I)} x^T A y_t + \frac{\alpha}{2}\|x - x'\|_2^2 + \frac{1}{2\eta}\|x - x_t\|_2^2, \\
y_{t+1} &= \arg\min_{y \in \Delta(J)} -x_t^T A y + \frac{\alpha}{2}\|y - y'\|_2^2 + \frac{1}{2\eta}\|y - y_t\|_2^2.
\end{aligned}
\tag{21}
$$

**Complexity:** (Theorem 3.4 and Corollary 3.5 of Sokota et al. (2022)) Let the squared error be $\varepsilon := \frac{1}{2}(\|x - x^*\|_2^2 + \|y - y^*\|_2^2)$. If $(x_t, y_t) \in \text{int } \Delta(I) \times \Delta(J)$ for all $t \in [T]$, and if $\eta$ is sufficiently small [2], then for an error threshold $\varepsilon_0 > 0$, $\varepsilon \leq \varepsilon_0$ if $T \geq \frac{\ln((I+J)/\varepsilon_0)}{\ln(1+\eta\alpha)}$. Thus MMD has complexity $\mathcal{O}(\ln((I + J)/\epsilon_0))$ with respect to the action space. **Remarks:** When $\alpha = 0$, MMD reduces to projected gradient descent ascent which is known to diverge or cycle for any positive learning rate. Sokota et al. (2022) showed empirically that MMD can be used to solve Nash by either annealing the amount of regularization over time or by having the magnet trail behind the current iterate. However, it is important to note that MMD assumes the solution to be interior, which is not the case in the games we consider when value is convex (no splitting) due to Isaacs' condition.

**FTRL variants** FTRL is a classic online learning algorithm known to converge in potential games but cycle in Hamiltonian games (Heliou et al., 2017; Mertikopoulos et al., 2018; Liu et al., 2024). To this end, variants of FTRL have been proposed to achieve last-iterate convergence to $\epsilon$-Nash or $\epsilon$-QRE (Perolat et al., 2021). Below we introduce a few of them to show that their complexities all increase with the size of the action space. **Algorithm:** *RegFTRL* (Liu et al., 2024) introduces regularization terms $(\phi_1, \phi_2)$ that are strictly convex and continuously differentiable on their respective simplex. For each iteration, do

$$
\begin{aligned}
x_{t+1} &= \arg\min_{x \in \Delta(I)} <x, \bar{y}_t> + \phi_1(x), & \bar{y}_t &= \sum_{\tau=1}^t A y_\tau + \alpha \nabla g_1(x_\tau), \\
y_{t+1} &= \arg\min_{y \in \Delta(J)} - <\bar{x}_t, y> + \phi_2(y), & \bar{x}_t &= \sum_{\tau=1}^t A^T x_\tau + \alpha \nabla g_2(y_\tau).
\end{aligned}
\tag{22}
$$

**Complexity:** *RegFTRL* is guaranteed to find an $\varepsilon$-QRE in $\mathcal{O}\left(\frac{\ln((I+J)/\varepsilon)}{\ln(1+\eta\alpha)}\right)$ iterations (Theorem 2 in Liu et al. (2024)). *FTRL-SP* (Abe et al., 2024) and *OMWU* (Rakhlin & Sridharan, 2013; Syrgkanis et al., 2015) finds $\varepsilon$-QRE in $\mathcal{O}\left(\frac{\ln((I+J)/\varepsilon)}{-\ln(1-\eta\alpha/2)}\right)$.

# D PROOFS FOR THE COMPLEXITY ANALYSIS OF CAMS

**Proof of Lem. 1**

---

[1] In Sokota et al. (2022), the authors used a more general regularization definition by introducing the Bregman divergence.

[2] See Corollary D.6 in Sokota et al. (2022) for details on the bound of $\eta$.

*Proof.* Note that $\sum_{k=1}^{I} \lambda^k = 1$. Then

$$\max_x |\tilde{V}(t, x, \tilde{z}) - V(t, x, z^*)| \le \max_x |\tilde{V}(t, x, \tilde{z}) - \tilde{V}(t, x, z^*)| + \max_x |\tilde{V}(t, x, z^*) - V(t, x, z^*)|$$

$$\le \epsilon_t^{minmax} + \max_x |\sum_{k=1}^{I} \lambda^k (\tilde{V}(t + \tau, x', p^k) - V(t + \tau, x', p^k))|$$

$$\le \epsilon_t^{minmax} + \epsilon_{t+\tau}^{bias}. \tag{23}$$

$\square$

**Proof of Lem. 2**

*Proof.*

$$\max_x |\hat{V}_t(x) - V(t, x)| \le \max_x |\hat{V}_t(x) - <\phi(x), w^*>| + \max_x |<\phi(x), w^*> - V(t, x)|$$

$$\le \max_x |<\theta^{\phi(x)}, \tilde{V}(t, x) - V(t, x)>| + \max_x |<\theta^{\phi(x)}, V(t, x) - \phi(x)^\top w^*>| + \epsilon_t^{app}$$

$$\le C(\epsilon_t^{minmax} + \epsilon_{t+\tau}^{bias} + \epsilon_t^{app}) + \epsilon_t^{app}. \tag{24}$$

$\square$

**Proof of Lem. 3**

*Proof.* Using Lem. 2 and by induction, we have

$$\epsilon_0^{bias} \le (\epsilon^{app} + C(\epsilon^{minmax} + \epsilon^{app}))\frac{1 - C^H}{1 - C} \le HC^H(\epsilon^{app} + C(\epsilon^{minmax} + \epsilon^{app})). \tag{25}$$

$\square$

**Proof of Thm. 1**

*Proof.* CAMS solves $TN$ minimax problems, each requires a worst-case $\mathcal{O}(\epsilon^{-4})$ iterations, and each iteration requires computing gradients of dimension $\mathcal{O}(I^2)$, considering the dimensionalities of action spaces as constants. This leads to a total complexity of $\mathcal{O}(TNI^2\epsilon^{-4})$. $\square$

# E    GAME SETTINGS, BASELINES AND GROUND TRUTH

## E.1    GAME SETTINGS

The players move in an arena bounded between $[-1, 1]$ in all directions. All games in the paper follow 2D/3D point dynamics as follows: $\dot{x}_j = Ax_j + Bu_j$, where $x_j$ is a vector of position and velocity and $u_j$ is the action for player $j$. Note that we use $u$ and $v$ in the optimization problems P$_1$ and P$_2$ to represent player 1 and player 2's actions respectively. The type independent effort loss for each player $j$ is defined as $l_j(u_j) = u_j^\top R_j u_j$, where $R_1 = diag(0.05, 0.025)$ and $R_2 = diag(0.05, 0.1)$. For the higher dimensional case, $R_1 = diag(0.05, 0.05, 0.025)$ and $R_2 = diag(0.05, 0.05, 0.1)$.

## E.2    GROUND TRUTH FOR HEXNER'S GAME

For the 4-stage and 10-stage Hexner's game, there exists analytical solution to the equilibrium policies via solving the HJB for respective players.

$$u_j = -R_j^{-1}B_j^\top K_j x_j + R_j^{-1}B_j^\top K_j \Phi_j z\tilde{\theta}_j,$$

based on the reformulation outlined below in which players' action $\tilde{\theta}_j \in \mathbb{R}$ become 1D and are decoupled from the state: where $\Phi_j$ is a state-transition matrix that solves $\dot{\Phi}_j = A_j\Phi_j$, with $\Phi_j(T)$ being an identity matrix, and $K_j$ is a solution to a continuous-time differential Ricatti equation:

$$\dot{K}_j = -A_j^\top K_j - K_j A_j + K_j^\top B_j R_j^{-1} B_j^\top K_j, \tag{26}$$

Finally, by defining

$$d_j = z^\top \Phi_j^\top K_j B_j R_j^{-1} B_j^\top K_j^\top \Phi_i z$$

and the critical time

$$t_r = \arg\min_t \int_0^t (d_1(s) - d_2(s))ds$$

and

$$\tilde{\theta}_j(t) = \begin{cases} 0, & t \in [0, t_r] \\ \theta, & t \in (t_r, T] \end{cases}.$$

As explained in Sec.6, P1 chooses $\theta_1 = 0$ until the critical time $t_r$ and P2 follows.

Note that in order to compute the ground truth when time is discretized with some $\tau$, we need the discrete counterpart of equation 26, namely the discrete-time Ricatti difference equation and compute the matrices $K$ recursively.

### E.3 OpenSpiel Implementations and Hyperparameters

We use OpenSpiel (Lanctot et al., 2019), a collection of various environments and algorithms for solving single and multi-agent games. We select OpenSpiel due to its ease of access and availability of wide range of algorithms. The first step is to write the game environment with simultaneous moves for the stage-game and the multi-stage games (with 4 decision nodes). Note that to learn the policy, the algorithms in OpenSpiel require conversion from simultaneous to sequential game, which can be done with a built-in method.

In the single-stage game, P1 has two information states representing his type, and P2 has only one information state (i.e., the starting position of the game which is fixed). In the case of the 4-stage game, the information state (or infostate) is a vector consisting of the P1's type (2-D: [0, 1] for type-1, [1, 0] for type-2), states of the players (8-D) and actions of the players at each time step ($4 \times 2 \times |\mathcal{A}|$). The 2-D "type" vector for P2 is populated with 0 as she has no access to P1's type. For example, the infostate at the final decision node for a type-1 P1 could be $[0, 1, x^{(8)}, \mathbb{1}_{u_0}^{(|\mathcal{A}|)}, \mathbb{1}_{v_0}^{(|\mathcal{A}|)}, \cdots, \mathbb{1}_{v_2}^{(|\mathcal{A}|)}, \mathbf{0}^{(|\mathcal{A}|)}, \mathbf{0}^{(|\mathcal{A}|)}]$, and $[0, 0, x^{(8)}, \mathbb{1}_{u_0}^{(|\mathcal{A}|)}, \mathbb{1}_{v_0}^{(|\mathcal{A}|)}, \cdots, \mathbb{1}_{v_2}^{(|\mathcal{A}|)}, \mathbf{0}^{(|\mathcal{A}|)}, \mathbf{0}^{(|\mathcal{A}|)}]$ for P2, where $u_k, v_k$ represent the index of the actions at $k^{th}$ decision node, $k = 0, 1, 2, 3$

The hyperparameters for DeepCFR is listed in table 2

Table 2: Hyperparameters for DeepCFR Training

| | |
|---|---|
| Policy Network Layers | (256, 256) |
| Advantage Network Layers | (256, 256) |
| Number of Iterations | 1000 (100, for $|\mathcal{A}| = 16$) |
| Number of Traversals | 5 (10, for $|\mathcal{A}| = 16$) |
| Learning Rate | 1e-3 |
| Advantage Network Batch Size | 1024 |
| Policy Network Batch Size | 10000 (5000 for $|\mathcal{A}| = 16$) |
| Memory Capacity | 1e7 (1e5 for $|\mathcal{A}| = 16$) |
| Advantage Network Train Steps | 1000 |
| Policy Network Train Steps | 5000 |
| Re-initialize Advantage Networks | True |

### E.4 VALUE NETWORK TRAINING DETAILS

**Data Sampling:** At each time-step, we first collect training data by solving the optimization problem ($P_1$ or $P_2$). Positions are sampled uniformly from [-1, 1] and velocities from $[-\bar{v}_t, \bar{v}_t]$ computed as $\bar{v}_t = t \times u_{max}$, where $u_{max}$ is the maximum acceleration. For the unconstrained game, $u_{max} = 12$ for both P1 and P2. For the constrained case, $u_{x_{max}} = 6$, $u_{y_{max}} = 12$ for P1 and $u_{x_{max}} = 6$, $u_{y_{max}} = 4$ for P2. During training, the velocities are normalized between [-1, 1]. The belief $p$ is then sampled uniformly from $[0, 1]$. For the dual value, we first determine the upper and lower bounds of $\hat{p}$ by computing the sub-gradient $\partial_p V(t_0, \cdot, \cdot)$ and then sample uniformly from $[\hat{p}^-, \hat{p}^+]$.

**Training:** We briefly discuss the training procedure of the value networks. As mentioned in the main paper, both the primal and the dual value functions are convex with respect to $p$ and $\hat{p}$ respectively. As a result, we use Input Convex Neural Networks (ICNN) (Amos et al., 2017) as the neural network architecture. Starting from $T - \tau$, solutions of the optimization problem $P_1$ for sampled $(X, p)$ is saved and the convex value network is fit to the saved training data. The model parameters are saved and are then used in the optimization step at $T - 2\tau$. This is repeated until the value function at $t = 0$ is fit. The inputs to the primal value network are the joint states containing position and velocities of the players $X$ and the belief $p$.

The process for training the dual value is similar to that of the primal value training. The inputs to the dual value network are the joint states containing position and velocities of the players $X$ and the dual variable $\hat{p}$.

## F DETAILS ON CONSTRAINED GAME

Here, we briefly explain the optimization problem for the constrained game. Formally, given the states $x_1, x_2$ of the players P1 and P2, the constraint is given by the function $c(x_1, x_2) = r - ||(p_{x_1}, p_{y_1}) - (p_{x_2}, p_{y_2})||_2$. P1 must always maintain a radial distance of $r$ from P2, else P1 receives a $+\infty$ penalty and P2 receives a reward of $-\infty$ (both want to minimize their costs).

We follow the method outlined in (Ghimire et al., 2024), and train a separate value function model $\mathcal{F} : t \times \mathcal{X} \to \mathbb{R}$, that classifies the state-space into safe ($\Omega_t$) and unsafe states. Safe states are those initial states from which P1 can avoid collision with P2, whereas unsafe states are those initial states from which it is impossible for P1 to avoid collision. The sub-zero level set of $\mathcal{F}$ correspond to the unsafe states.

With $\mathcal{F}$ available, we can query it to check if the resulting states $x^k = \text{ODE}(x, \tau, u^k, v^k; f)$ in equation $P_1$ are unsafe. If so, a high penalty is added (subtracted in the case of dual game), otherwise 0. Formally, at some time-step $t$, initial state $x \in \Omega_t$, $p$ (resp. $\hat{p}$), we solve the following optimization problem for the constrained primal ($V$) (resp. dual ($V^*$)) value:

$$
\min_{\{u^k\},\{\alpha_{ki}\}} \max_{\{v^k\}} \sum_{k=1}^{I} \lambda^k \left( V(t+\tau, x^k, p^k) + \tau\mathbb{E}_{i\sim p^k}[l_i(u^k, v^k)] \right) + \gamma \cdot \text{relu}(-\mathcal{F}(t+\tau, x^k)
$$

$$
\text{s.t.} \quad u^k \in \mathcal{U}, \quad x^k = \text{ODE}(x, \tau, u^k, v^k; f), \quad v^k \in \mathcal{V}, \quad \alpha_{ki} \in [0, 1],
$$

$$
\sum_{k=1}^{I} \alpha_{ki} = 1, \quad \lambda^k = \sum_{i=1}^{I} \alpha_{ki}p[i], \quad p^k[i] = \frac{\alpha_{ki}p[i]}{\lambda^k}, \quad \forall i, k \in [I].
$$

(27)

$$
\min_{\{v^k\},\{\lambda^k\},\{\hat{p}^k\}} \max_{\{u^k\}} \sum_{k=1}^{I+1} \lambda^k \left( V^*(t+\tau, x^k, \hat{p}^k - \tau l(u^k, v^k)) \right) - \gamma \cdot \text{relu}(-\mathcal{F}(t+\tau, x^k)
$$

$$
\text{s.t.} \quad u^k \in \mathcal{U}, \quad v^k \in \mathcal{V}, \quad x^k = \text{ODE}(x, \tau, u^k, v^k; f), \quad \lambda^k \in [0, 1], \quad (28)
$$

$$
\sum_{k=1}^{I+1} \lambda^k \hat{p}^k = \hat{p}, \quad \sum_{k=1}^{I+1} \lambda^k = 1, \quad k \in [I+1].
$$

where $\gamma$ is a scaling factor.

