# OpenReview forum: "Two-Player Zero-Sum Differential Games with One-Sided Information and Continuous Actions"
_ICLR.cc/2025/Conference — Submitted to ICLR 2025_

### Official Review · Reviewer_Zbcv · 2024-11-01

**Soundness:** 1
**Presentation:** 1
**Contribution:** 1
**Rating:** 3
**Confidence:** 4

**Summary:**

In this work, the authors consider the problem of solving a differential two-player zero-sum game with one-sided information, meaning a game where only one player is informed about the "type" of the terminal function, i.e., the reward/loss function.

**Strengths:**

Unfortunately, I cannot identify any significant strengths in this work. The writing is so unclear that it is nearly impossible to follow.

**Weaknesses:**

Immediately upon beginning to read the paper, it becomes evident that it requires extensive rewriting. The phrasing is often either ambiguous or nonsensical. This issue is apparent even from the abstract, where the authors attempt to present their results with phrases like "a computational complexity independent of $|A|$ can be achieved," when they actually mean "a method/algorithm with computational complexity independent of $|A|$ can be achieved." They define CAMS as approximating continuous-action mixed strategies, but then ambiguously state “we demonstrate correctness of CAMS.”

Throughout the paper, there are missing articles and numerous errors. Additionally, the definition of zero-sum differential games is presented hastily; understanding the concept requires consulting other sources. Even the statement of the problem—that in zero-sum differential games, P1 seeks to minimize the terminal payoff function $g_i(x(T))$ (plus a running cost) while P2 aims to maximize it—is not clearly stated. The equilibrium, which is presumably the key concept, is not defined. Even in the introduction, the authors mention a no-regret algorithm without explaining what no-regret is. They reference no-regret algorithms, but the work instead presents a direct game-solving method, rather than focusing on minimizing regret.

Furthermore, in the introduction, the authors claim to reformulate a convex-concave problem (which is not clearly articulated) into a nonconvex-nonconcave one (also not clearly articulated) that they assert can be solved easily, though without clear reasoning. Overall, there are missing explanations throughout, making it nearly impossible for the reader to follow. Here are a few additional specific examples:

- **Line 128**: The motivation for including the additional running cost $l(u,v)$ is unclear.
- **Line 130**: The phrase “has a value” is misattributed, and the concept of "the value" is only explained later on the page.
- **Line 143**: "to have pure Nash" lacks clarity and does not seem adequately explained.
- **Line 175**: In equation 5, the expected payoff includes $V_\tau$ and $l(u,v)$, though the former represents the expected value of the terminal payoff plus the integral of the running cost $l(u,v)$.
- **Line 179**: The phrase “has a pure equilibrium” is not defined.
- **Line 179**: The sentence “If $U_\tau$ is not convex in $p$, P1…” is nonsensical.

The authors need to introduce the problem more clearly, describe its inherent difficulties, and explain how and why their proposed method is able to overcome these difficulties.

**Questions:**

Please read the comments above.

---

> ### Author Response · Authors · 2024-11-20
> **Response to Reviewer Zbcv**
>
> We appreciate the time and effort the reviewer has taken to provide feedback on our submission. The points raised are invaluable in highlighting areas that require clarification and further elaboration.
>
> We would like to respectfully highlight some of the reviewer’s comments that may have resulted from misunderstandings or misinterpretations:
>
> - The reviewer pointed out that the phrase “a computational complexity independent of $|\mathcal{A}|$ can be achieved” is ambiguous. We agree that this phrasing could be improved for clarity. However, we believe the intent was clear in the context of describing our algorithm. We will revise the phrasing as outlined by the reviewer to avoid ambiguity.
>
> - The reviewer mentioned the ambiguity in the phrase “we demonstrate correctness of CAMS”. To clarify, by “correctness” we mean that we validate the effectiveness of the proposed algorithm by comparing the resultant approximations of the equilibrium strategies with the ground truth. All algorithms for solving Bellman or Hamilton-Jacobi equations resort to value/policy approximation. Our algorithm, just like others (e.g., CFR and MMD), is correct in the sense that its approximation approaches the true solution when the time interval approaches 0+ [1] and the number of subgames we solve goes to infinity. Importantly, our algorithm is superior than CFR and MMD for games with continuous action spaces because it has a computational complexity independent from $|\mathcal{A}|$.
>
> > **Even the statement of the problem—that in zero-sum differential games, P1 seeks to minimize the terminal payoff function
> $g_i(x(T))$ (plus a running cost) while P2 aims to maximize it—is not clearly stated.**
>
> On page 3 line 128, we introduced the terminal and instantaneous payoffs at the beginning of the formal definition of the game. On line 157, we made it clear that P1 minimizes the cost and P2 maximizes. We could make this setting more explicit, but one reason for not highlighting this particular setting is because it does not affect the solution structure we reveal in this paper.
>
> > **The equilibrium, which is presumably the key concept, is not defined.**
>
> This is actually done on page 3 line 129: “We introduce the following assumptions under which the game has a value”. And then Eq. (3) defines the value and equilibrium strategies formally. We acknowledge that our introduction of the equilibrium is lean and may not be standard for the EFG community. This can be readily improved in the revision.
>
> > **Even in the introduction, the authors mention a no-regret algorithm without explaining what no-regret is.**
>
> Due to limited space, we have to assume that our audience has familiarity with the context, which seems to be the case.
>
> > **They reference no-regret algorithms, but the work instead presents a direct game-solving method, rather than focusing on minimizing regret.**
>
> We discuss no-regret algorithms because of their empirical efficiency and widespread applications to solving incomplete and imperfect information games. Our intention is to highlight the limitations of these algorithms when applied to differential games with continuous action spaces.
>
> In contrast, our work presents a direct game-solving method that efficiently handles continuous action spaces without relying on discretization. By doing so, we overcome the inefficiencies associated with existing no-regret algorithms, providing a scalable solution for computing equilibrium strategies of differential games. This distinction underscores the novelty and practical advantages of our approach in the context of differential games.
>
> [1] Cardaliaguet, Pierre. "Numerical approximation and optimal strategies for differential games with lack of information on one side."

---

> ### Author Response · Authors · 2024-11-20
> **Response to Reviewer Zbcv (contd.)**
>
> > **the authors claim to reformulate a convex-concave problem (which is not clearly articulated) into a nonconvex-nonconcave one (also not clearly articulated) that they assert can be solved easily, though without clear reasoning.**
>
> Our statement regarding solving the nonconvex-nonconcave problem is “(if it is) of a much smaller scale (then it) becomes more efficient to solve **in practice**”. This is different from the reviewer’s rephrasing “(it) can be solved easily”. To be more precise, IIEFG solver considers the behavioral strategy pair at each subgame the solution of a minimax problem, which scales with the size of the action space, $|\mathcal{A}|$. Since strategies are considered to be distributions over the actions, the problem has a bilinear form and is thus convex-concave. We propose to reformulate this problem by exploiting the splitting structure of the equilibrium (discussed on Page 4 Behavioral strategy of P1). This leads to a minimax problem of a much smaller scale (independent from $|\mathcal{A}|$) and without action discretization, yet is nonconvex-nonconcave in general (see Sec. 4 Reformulation of the primal and dual games). Directly comparing the computational complexities for solving the original minimax problem and its reformulation is difficult. Therefore, we justify the advantage of our solver through empirical studies (Sec. 6).
>
> > **Line 128: The motivation for including the additional running cost is unclear.**
>
> Running costs commonly exist in both differential games and POMG/EFGs. For differential games defined on dynamical systems such as aircraft, human, robot, etc., the running cost may define fuel/energy consumption, system stability, safety penalties, etc.
>
> > **Line 130: The phrase “has a value” is misattributed, and the concept of "the value" is only explained later on the page.**
>
> We apologize for the confusion. We can reorganize the introduction to make it more friendly for the general audience.
>
> > **Line 143: "to have pure Nash" lacks clarity and does not seem adequately explained.**
>
> “Pure Nash” here means a Nash equilibrium in the pure strategy space.
>
> > **Line 175: In equation 5, the expected payoff includes $V_\tau$ and $l(u, v)$, though the former represents the expected value of the terminal payoff plus the integral of the running cost $l(u, v)$**
>
> Equation (5) is the bellman-backup to compute the (non-revealing) value at some state $(t, x, p)$, while $V(t_0, x_0, p)$ is the value of the game starting from $(x_0, p)$. We can always re-write the value as “the running cost of going from $x$ to $x’$ in some interval $\tau$)” + “$V(t+\tau, x’, p)$: the future equilibrium value from $x’$ onward”. Please let us know if this helps.
>
> > **Line 179: The sentence “If $U_\tau$ is not convex in $p$, P1…” is nonsensical.**
>
> We will rephrase this sentence to convey the following idea: P1 can achieve a better outcome by convexifying $U_\tau$ through the use of a mixed strategy if $U_\tau$ is not convex in $p$.

---

> > ### Comment · Reviewer_Zbcv · 2024-11-26
> >
> > I appreciate the author’s efforts and the time spent addressing the points I raised. However, this work contains abundant confusing parts, lacks formal definitions, and is missing a coherent narrative. While some parts could be improved during revision, I believe it requires substantial rewriting to meet the standards of the conference.

---

> > > ### Author Response · Authors · 2024-11-27
> > >
> > > Thank you for your continued engagement. We would greatly appreciate it if you could confirm whether our explanations have sufficiently addressed your previous concerns. If there are any areas that remain unclear or need further attention, we would welcome your guidance to resolve them. Alternatively, if our responses have clarified your earlier concerns, we would be grateful to know if there are additional issues that might justify your current score.

---

### Official Review · Reviewer_CzPX · 2024-11-03

**Soundness:** 1
**Presentation:** 1
**Contribution:** 2
**Rating:** 3
**Confidence:** 2

**Summary:**

The paper studies zero-sum two player differentiable games with one-side information and infinite action space.
The paper shows that under some conditions (Cav u and Isaacs conditions) there exists an algorithm with complexity that does not depend on the cardinality of the action space. Experimental evaluation is provided

**Strengths:**

Finding algorithms with better dependency on the cardinality of the action space is an important topic.

**Weaknesses:**

The paper has important presentation issues. Someone non familiar with differentiable games has difficulty even understanding the setup.
Moreover the authors do not make a great job at differentiating between their contributions and known results. In the reviewer's opinion the paper needs substantial rewriting.
For example, from section 3, it seems that the game is in continuous time, while the related works section speaks about cfr+ and deep CFR, which are designed for EFG. What is the exact setting considered by the paper? The theoretical grounding of the works seems to be based on Cardaliaguet (2007), which is based on different assumptions/language/theory than the works the authors cite as related works. This might be a strong point in favor of the paper (bringing concepts from another field into the computational game theory framework); however, the authors should invest more effort in relating the two.
I even struggle to understand what the Cav u property is.

**Questions:**

Have you considered the online convex optimization point of view of learning in EFG? There, you consider a continuous actions space (although a polytope). More in general, games with continuous actions are not unfeasible to optimize per se. How does this relate?
What exactly are your technical contributions? Theorem 1 and related lemmas seem straightforward from existing results.
Is the game continuous or discrete time? How can you adapt something like CFR to work in continuous time?

---

> ### Author Response · Authors · 2024-11-20
> **Response to Reviewer CzPX**
>
> We would like to thank the reviewer for the comments and questions. Here we try to address your concerns and hope for continued fruitful discussions.
>
> We focus on differential games, which are continuous-time and have continuous state and action spaces. That being said, computationally tractable algorithms for solving differential games often consider a discrete-time formulation (as in Eq. (6) for the primal game and Eq. (9) for the dual game). The validity of this discretization is supported by the existing proof that value and policy approximation through the discrete-time formulation uniformly converges to the continuous-time solution as the discrete time interval approaches 0+ [1]. Therefore, when it comes to algorithm design, both differential games and IIEFGs consider discrete time. If we further discretize the action space, IIEFG solvers such as CFR and MMD can be applied to our problem yet are inefficient since the computational complexities of such algorithms grow along the size of the discretized action set.
>
> As the reviewer pointed out, one of the intended contributions of this paper is indeed to connect computational game theory–which has largely focused on games with discrete action sets (other than PSRO type of algorithms that lack rigorous complexity analyses [2, 3]) and differential game theory, which has largely been limited to theoretical value existence proofs without scalable solvers. The difference in language between the two has been a major obstacle in associating these mathematically related games.
>
> We acknowledge that the presentation of some concepts may be dense, especially for readers not extremely familiar with differential games. We have made significant efforts to make the paper accessible, but we understand that the inherent complexity of the subject matter may still pose challenges. We appreciate your patience and hope that, with careful consideration, the connections and contributions we present will become clearer.
>
> > **Q1: Have you considered the online convex optimization point of view of learning in EFG? There, you consider a continuous actions space (although a polytope). More in general, games with continuous actions are not unfeasible to optimize per se. How does this relate? What exactly are your technical contributions?**
>
> Yes, we are familiar with online learning since some IIEFG solvers essentially use paired online learners [4, 5, 6], in which case the convex search space is the simplex that defines distributions over a finite set of actions.
>
> In our paper, we discuss that while these solvers are effective for games with discrete action sets, their computational complexity scales with the size of the action space. This scaling becomes prohibitive when there is a need to discretize a continuous action space finely enough to capture the necessary strategic nuances, making IIEFG solvers inefficient for this setting.
>
> Regarding the comment on “continuous action spaces are not inherently unfeasible to optimize”. We would like to note that our focus is not on optimizing over pure strategies (sequences of continuous actions) but rather mixed strategies (distributions over the pure strategy space), since Nash equilibrium is defined on mixed strategies for incomplete-information games. To our knowledge, existing online learners do not solve for distributions over continuous convex sets.
>
> Our key contribution is in explaining the fact that for 2p0s differential games with incomplete information and Isaacs’ condition, instead of solving for distributions over action spaces at all infostates, these distributions can only take two forms: They either collapse to a single action when the non-revealing value (U in Eq. (5)) is convex due to Isaacs’ condition, or being atomic on at most I actions when U is nonconvex, where I is the number of possible payoffs. Essentially, the game tree can be trimmed so that every node (infostate) has at most I branches, rather than an infinite number of branches due to the continuous action space. This insight makes the complexity of our algorithm independent from the size of the action space.
>
> [1] Cardaliaguet, Pierre. (2009). "Numerical approximation and optimal strategies for differential games with lack of information on one side."
>
> [2] Lanctot, Marc, et al. (2017). "A unified game-theoretic approach to multiagent reinforcement learning."
>
> [3] McAleer, Stephen, et al. (2021). "XDO: A double oracle algorithm for extensive-form games."
>
> [4] McMahan, B.. (2011). "Follow-the-Regularized-Leader and Mirror Descent: Equivalence Theorems and L1 Regularization."
>
> [5] Perolat, Julien, et al. (2021). "From poincaré recurrence to convergence in imperfect information games: Finding equilibrium via regularization."
>
> [6] Farina, Gabriele, et al. (2019). "Online convex optimization for sequential decision processes and extensive-form games."

---

> > ### Author Response · Authors · 2024-11-20
> > **Response to Reviewer CzPX (contd.)**
> >
> > > **Q2: Is the game continuous or discrete time? How can you adapt something like CFR to work in continuous time?**
> >
> > As we discussed above, the game occurs in continuous time, but like other approximation methods, our algorithm solves its discrete counterpart as we mentioned on line 170 (“..we first introduce the discrete-time Bellman backup for approximating the solution..”). We will mention this explicitly in the revision. If we discretize both time and the action space, IIEFG solvers such as CFR and MMD can be applied.

---

> > > ### Comment · Reviewer_CzPX · 2024-11-26
> > >
> > > I want to thank the authors for taking the time to answer my questions. I still think the paper needs substantial rewriting, thus I'll keep my score.

---

> > > > ### Author Response · Authors · 2024-11-27
> > > >
> > > > Thank you for your feedback and for considering our responses. As we aim to use this discussion period productively, we kindly ask if our rebuttals have adequately addressed the concerns you raised earlier. If there are remaining issues or specific aspects of the paper that you believe could be improved, we would appreciate further details to help us focus our efforts effectively.

---

### Official Review · Reviewer_Q83L · 2024-11-04

**Soundness:** 3
**Presentation:** 2
**Contribution:** 3
**Rating:** 6
**Confidence:** 3

**Summary:**

This paper proposes an algorithm for solving two player zero sum differential games with one-sided information under the assumption that Isaac's condition holds. In particular, the paper exploits the fact that in a one-sided imperfect information game with a finite set $I$ of payoff types and  the imperfect information is on the payoff types, the game (which originally could have an infinitely large action set) can be reformulated as a a nonconvex-nonconcave problem of size $O(I^2)$, which can potentially be much smaller than the original game size. Then, the paper introduces an algorithm called Continuous Action Mixed Strategy solver that is able to achieve approximate Nash equilibria of the game with a complexity of $O(TNI^2 \epsilon^{-4})$. The key contribution of this paper is the characterization of a class of differential games for which tractable solutions can be found even with continuous action spaces, and the accompanying analysis of an algorithm which exploits the structure of the game to compute equilibria faster than SOTA methods.

**Strengths:**

- To my knowledge, this is the first paper that gives a framework to solve 2p0s differential games where the imperfect information is specifically about the payoff types and is one-sided in the sense that P1 obtains information about the payoff type before P2. While this seems like a fair niche class, the paper gives game examples from the literature that satisfy this condition, showing that the proposed algorithm performs well in practice and indeed shows time-invariance to increasing action sets since the underlying payoff types are invariant.
- The paper gives a thorough description and analysis of the specific class of games studied and the accompanying primal-dual reformulation. Overall a good description of why and when the proposed approach works is given, and the structure of the paper as a whole is good.

**Weaknesses:**

- The paper gives a comprehensive analysis which crucially hinges on the assumptions of information asymmetry and a finite set of payoff types. However, there is a lack of explanation for when these assumptions are natural as well as when or how one can verify that a game lives in this class. It seems that Hexner's game is certainly an important game in this setting, but are there more game examples that can be shown to satisfy these assumptions?
- There are a few typos in the paper (some are listed below), and the notation especially in the preliminaries are quite dense. The paper could be improved by taking more time/space to explain the notation (particularly for readers unfamiliar with differential games), perhaps in the appendix.

Typos:
- References for Diplomacy seem to be broken, displaying as (, FAIR).
- 'tractable solve' in the last line of the conclusion is a grammatical error.
- Some typos in the references, e.g. P Cardaliaguet vs Pierre Cardaliaguet; some proper nouns are not properly capitalized.

**Questions:**

- A question about the experiments on Hexner's game: Unless I am misunderstanding the results, the normal-form variants of Hexner's game you experimented with are all with finite $\vert A \vert$. Would there be an easy way to verify your results in a game where $\vert A \vert = \infty$ but $I$ is finite?
- Given that the games you study are of the pursuit-evasion type, would it also be interesting to study Stackelberg equilibria and algorithms to compute Stackelberg equilibria in this setup?

---

> ### Author Response · Authors · 2024-11-20
> **Response to Reviewer Q83L**
>
> We thank the reviewer for taking the time to thoroughly review our work and for the constructive feedback. Kindly find our responses to your questions and concerns below:
>
> > **Are there more game examples that can be shown to satisfy these assumptions?**
>
> The games we study use the following assumptions: (1) there is a finite set of possible payoffs, (2) Isaacs’ condition holds, and (3) payoff information is one-sided. We argue that there are abundant games satisfying these assumptions with reasonable abstraction from real-world interactions such as competitive sports (e.g., penalty/corner kicks in soccer) and defense applications (e.g., missile defense):
>
> 1. While real-world applications may have continuous payoff sets (e.g., the whole goal area for a penalty taker), it is common practice to use discretization (e.g., four corners of the goal) to facilitate the derivation of prescriptive solutions.
> 2. A sufficient condition for the Isaacs’ condition to hold is when the dynamical system is control-affine. This is a common assumption in studies on robust safety/stability of dynamical systems (e.g., cars, drones, and robots).
> 3. While both players may hold private information, it is reasonable in many applications to use a conservative model of the game from the defender’s perspective by assuming that the attacker has full information, e.g., missile defense. Solving the one-sided game is also a necessary first step since scalable solvers have not yet been developed for differential games with either one- or two-sided information.
>
> > **There are a few typos in the paper (some are listed below), and the notation especially in the preliminaries are quite dense.**
>
> We agree that the notation is quite dense and made an effort to simplify as much as possible without hindering the structure of the paper. We shall further refine the writing so that the introduction is concise, coherent, and clear.
>
> > **Q1: A question about the experiments on Hexner's game: Unless I am misunderstanding the results, the normal-form variants of Hexner's game you experimented with are all with finite A. Would there be an easy way to verify your results in a game where |A|=∞ but I is finite?**
>
> Hexner’s game, including its normal-form version, has a continuous action space, so it is indeed a game with $|\mathcal{A}| = \infty$ but finite $I$. It is also the only game of this type we have found that has an analytical Nash equilibrium. However, we use a range of discrete action spaces when solving this game using IIEFG solvers (CFR+ and MMD), since these solvers are designed for discrete action spaces. Our algorithm (CAMS), on the other hand, solves the game without action discretization.
>
> > **Q2: Given that the games you study are of the pursuit-evasion type, would it also be interesting to study Stackelberg equilibria and algorithms to compute Stackelberg equilibria in this setup?**
>
> It would indeed be interesting to study Stackelberg equilibria in the context of our framework. However, our current work focuses on differential games with simultaneous moves and one-sided information, and the existing theoretical framework is primarily tailored to this setting.
> Just as a mental exercise, we have the following thoughts without proof. If “Stackelberg” means that P1 moves first and P2 next, then the Nash equilibrium is captured by the primal game (Eq. (6)), since P2 can always play the (pure) best response to P1’s actions. If P2 moves first and P1 next, the Nash equilibrium is captured by the dual game (Eq. (9)), since P1 can play the best responses to P2’s actions.

---

> > ### Comment · Reviewer_Q83L · 2024-11-27
> > **Response to rebuttal**
> >
> > Thank you for your response. I think if rewriting/cleaning up of notation is done the paper will definitely be more readable, so I am happy to maintain my score.

---

### Author Response · Authors · 2024-11-25

We sincerely appreciate the time and effort all reviewers have invested in evaluating our work. Their diverse perspectives have provided valuable insights that will help us improve our paper.

We are particularly encouraged by the positive feedback from Reviewer **Q83L**, who recognized our work as the **“first paper that gives a framework for 2p0s differential games where the imperfect information is specifically about the payoff types and is one-sided”**. They also commended the paper’s **“thorough description and analysis”** and its **“good description of why and when the proposed approach works”**. Additionally, Reviewer **CzPX** noted that **“finding algorithms with better dependency on the cardinality of the action space is an important topic,”** which aligns with our core motivation and contribution.

We acknowledge that our paper addresses a complex topic at the intersection of differential game theory and computational game theory. This interdisciplinary nature may have led to some challenges in comprehension, especially for those more familiar with one field than the other. We take full responsibility for any lack of clarity and are committed to improving the presentation to make our work more accessible to a broader audience.

We recognize the concerns raised by Reviewers **CzPX** and **Zbcv** regarding clarity and presentation. In our rebuttals, we have addressed these issues, including clarifications on the game setting, the definition of equilibrium, and the motivation behind our approach. We have also explained how our work connects differential games with computational game theory methods, and how our algorithm overcomes limitations of existing solvers when applied to games with continuous action spaces. We believe these clarifications resolve many of the reviewers’ concerns and further highlight the novelty and practical benefits of our approach.

---

### Meta-Review · Area_Chair_g1TT · 2024-12-20

**Metareview:**

The paper shows that efficient equilibrium computation is possible for a class of asymmetric-information differential games that satisfies certain technical conditions.

All reviewers had a background on learning and equilibrium computation in games. While they seemed interested in the contribution, there was consensus that the paper's presentation was confusing and not enough self-contained. These difficulties likely prevented the reviewers from fully appreciating whether the technical claims of the paper are correct. With only one reviewer weakly in support of the paper, and two reviewers confident that the paper should be rejected, a rejection decision was made.

**Additional Comments On Reviewer Discussion:**

The discussion mostly revolved around clarying certain parts of the writeup and some connections to existing work. In general, the reviewers found the responses helpful. However, the reviewers felt that even with the given clarifications, the paper requires substantial work before it can be accepted.

---

### Decision · Program_Chairs · 2025-01-22

Reject